

# Coherent forward scattering as a robust probe of multifractality in critical disordered media

Maxime Martinez[1], Gabriel Lemarié[1,2,3], Bertrand Georgeot[1],
Christian Miniatura[2,3,4,5,6] and Olivier Giraud[7]

**1** Laboratoire de Physique Théorique, Université de Toulouse, CNRS, UPS, France
**2** MajuLab, CNRS-UCA-SU-NUS-NTU International Joint Research Unit, Singapore
**3** Centre for Quantum Technologies, National University of Singapore, Singapore
**4** Université Côte d'Azur, CNRS, INPHYNI, Nice, France
**5** Department of Physics, National University of Singapore, Singapore
**6** School of Physical and Mathematical Sciences,
Nanyang Technological University, Singapore
**7** Université Paris Saclay, CNRS, LPTMS, 91405 Orsay, France

## Abstract

We study coherent forward scattering (CFS) in critical disordered systems, whose eigenstates are multifractals. We give general and simple arguments that make it possible to fully characterize the dynamics of the shape and height of the CFS peak. We show that the dynamics is governed by multifractal dimensions $D_1$ and $D_2$, which suggests that CFS could be used as an experimental probe for quantum multifractality. Our predictions are universal and numerically verified in three paradigmatic models of quantum multifractality: Power-law Random Banded Matrices (PRBM), the Ruijsenaars-Schneider ensembles (RS), and the three-dimensional kicked-rotor (3DKR). In the strong multifractal regime, we show analytically that these universal predictions exactly coincide with results from standard perturbation theory applied to the PRBM and RS models.



# 1   Introduction

Wave transport in disordered systems is a long-standing topic of interest in mesoscopic physics. In particular, wave interference can have dramatic consequences on quantum transport properties. The most celebrated example is probably Anderson localization (AL) [1], that is, the suppression of quantum diffusion and the exponential localization of quantum states. AL is ubiquitous in wave physics and has been observed in many experimental situations: with acoustic waves [2,3], light [4–8], matter waves [9–15].

Appearance of AL depends on several characteristics, in particular dimensionality, disorder strength and correlations. For instance, it is well established that 3d disordered lattices undergo a genuine disorder-driven metal-insulator transition (MIT), associated with a mobility edge in the spectrum, separating the insulating phase with localized eigenstates from the conducting phase with extended eigenstates. Near the critical point of such disorder driven transitions, eigenstates $\phi_\alpha$ (with energy $\omega_\alpha$) can display multifractal behavior, for instance at the MIT in Anderson model [16–18] and graphs [19–21], but also for Weyl-semimetal–diffusive transition [22]. They are extended but non-ergodic, and characterized by the anomalous scaling of their moments $I_q(E)$:

$$I_q(E) = \frac{\langle \sum_{\mathbf{n},\alpha} |\phi_\alpha(\mathbf{n})|^{2q} \delta(E - \omega_\alpha) \rangle}{\langle \sum_\alpha \delta(E - \omega_\alpha) \rangle} \sim N^{-D_q(q-1)}, \tag{1}$$

where $D_q$ are the multifractal dimensions, forming a continuous set with $q$ real ($\langle \ldots \rangle$ represents an average over disorder configurations). Extreme cases $D_q = 0$ and $D_q = d$ (the dimension of the system) for all $q$, correspond respectively to localized and extended ergodic eigenstates.

While Anderson MIT has been observed directly in atomic matter waves [13], experimental observation of multifractality remains challenging [23–26]. In particular, there exists to our knowledge no direct experimental observation of dynamical multifractality, i.e. manifestation of multifractality through transport properties (e.g. power-law decay of the return probability [27, 28]).

Another celebrated wave interference effect is the coherent backscattering (CBS). It describes the doubling of the scattering probability (with respect to incoherent classical contribution) of an incident plane wave with wave vector $\mathbf{k}_0$, in the backward direction $-\mathbf{k}_0$. Coherent backscattering has been observed in many experimental situations: with light [29–33], acoustic waves [34,35], seismic waves [36] and cold atoms [37,38]. Recently, it was demonstrated that in the presence of AL a new robust scattering effect emerges [39–46], namely the doubling of the scattering probability in the *forward* direction $+\mathbf{k}_0$. This phenomenon, which appears at long times, was dubbed *coherent forward scattering* (CFS). CBS and CFS actually have a distinct origin: CBS comes from pair interference of time-reversed paths (and thus requires time-reversal symmetry), while CFS is present even in the absence of time-reversal symmetry [39,40]. From an experimental point of view, CFS has recently been observed with cold atoms [38].

In this work, we discuss the fate of CFS at the critical point of a disorder-driven transition with multifractal eigenstates. This problem was first addressed for a bulk 3d Anderson

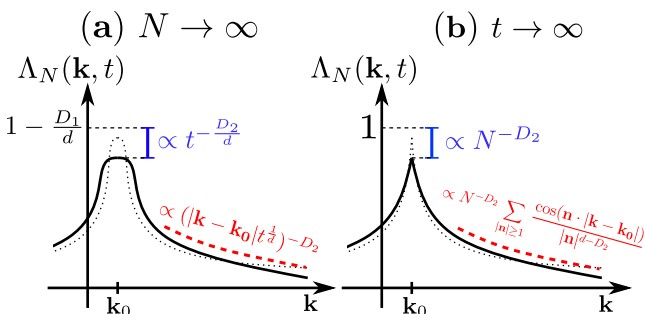

Figure 1: CFS contrast $\Lambda_N(\mathbf{k}, t; E)$ defined by (40) in critical disordered systems. $\mathbf{k}_0$ is the wave vector of the incident plane wave and $D_q$ are the multifractal dimensions of the eigenstates. (a) In systems of infinite size $N \to \infty$, the emergence of the CFS peak as a function of time is governed by the nonergodic properties of multifractal eigenstates. The CFS wings decay asymptotically like $(|\mathbf{k} - \mathbf{k}_0| t^{1/d})^{-D_2}$, see Eqs. (62) and (63), while the CFS peak height grows algebraically in time like $t^{-D_2/d}$ and finally reaches the compressibility value $\chi = 1 - \frac{D_1}{d}$ in the long-time limit $t \to \infty$, see Eq. (58). (b) For systems of finite size $N$, the long-time dynamics of the CFS peak is governed by the box boundaries. The CFS peak height reaches $1 - \alpha N^{-D_2}$ for $t \to \infty$ with $\alpha$ some numerical factor, see Eq. (56). The wings of the CFS peak are then described by Eq. (54).

lattice [44], for which it was shown that CFS survives at the transition, with however a scattering probability smaller than in the localized phase. More precisely, it was conjectured from numerical evidence that, instead of a doubling of the classical incoherent contribution, the forward scattering probability corresponds to a multiplication by a factor $(2 - D_1/d)$, with $d$ the dimension of the system and $D_1$ the information dimension. In our previous study [46], we gave scaling arguments that corroborate this conjecture, backed by numerical simulations on the Ruijsenaars-Schneider ensemble, a Floquet system with critical disorder and tunable multifractal dimensions. We also studied CFS at the transition in finite-size systems, unveiling a new regime, where CFS properties have finite-size scaling related to the multifractal dimension $D_2$ [46].

This article is based on the approach developed in our previous work [46] and, somehow, in the spirit of the random matrix theory point of view discussed in [41]. In particular, we give a complete description of the dynamics of CFS peak in critical disordered systems, including height and shape of the scattering probability, in two distinct dynamical regimes. Our findings are summarized in the sketch in Fig. 1. In particular, we present new links between CFS dynamics and the multifractal dimension $D_2$, that are relevant for most experimental situations. Our analytical predictions are verified on three different critical disordered models with multifractal eigenstates: Power-Law Random Banded Matrices (PRBM), Ruijsenaars-Schneider ensemble (RS) and unitary three-dimensional random Kicked Rotor (3DKR). Our predictions are also corroborated by perturbative expansions for RS and PRBM models in the strong multifractality regime. These results pave the way to a direct observation of a dynamical manifestation of multifractality in a critical disordered system.

## 2 Critical disordered models

As explained, in the following, our predictions will be compared to numerical simulations on three different models. All of them can be mapped onto the generalized $d$-dimensional

Table 1: Summary of some of the main properties of the three models considered in this article (see text for more details).

| Model | PRBM | RS | 3DKR |
|---|---|---|---|
| Tunable multifractal dimensions $D_q$ | Yes with $b \in [0, \infty[$ | Yes with $a \in [0, \infty[$ | No |
| Type | Hamiltonian | Floquet | Floquet |
| Energy dependent properties | Yes | No | No |
| Hopping range $t_n$ | Long-range $\sim 1/n$ | Long-range $\sim 1/n$ | Short-range (exponential decay) |
| Dimension | $d = 1$ | $d = 1$ | $d = 3$ |
| Direct (disorder) space | Position | Momentum | Momentum |

Anderson model, defined by the following tight-binding Hamiltonian

$$\hat{H} = \sum_{\mathbf{n}} \varepsilon_{\mathbf{n}} |\mathbf{n}\rangle\langle\mathbf{n}| + \sum_{\mathbf{n}\neq\mathbf{m}} t_{\mathbf{nm}} |\mathbf{n}\rangle\langle\mathbf{m}| \,, \tag{2}$$

where $|\mathbf{n}\rangle$ are the lattice site states, $\varepsilon_{\mathbf{n}}$ the on-site energies and $t_{\mathbf{nm}}$ the hopping between two sites at distance $|\mathbf{n}-\mathbf{m}|$. Both $\varepsilon_{\mathbf{n}}$ and $t_{\mathbf{nm}}$ can be considered arbitrary random variables, whose exact properties will depend on the system considered (see Table 1). We will be interested in finite-size effects, and will consider a system with linear size $N$, i.e. with a total number of sites equal to $N^d$.

MIT in the generalized Anderson model (2) has been intensively studied (see [18, 47] and references therein). The three relevant parameters are the spatial dimension $d$ of the lattice, the range of the hopping $t_{\mathbf{nm}}$, and existence of correlations in the random entries of the Hamiltonian. We recall here some well established facts: (i) in the absence of disorder correlations and if $\langle|t_{\mathbf{nm}}|\rangle$ decay faster than $1/|\mathbf{n}-\mathbf{m}|^d$, Anderson transition only occurs for $d > 2$; (ii) in the absence of disorder correlations, critical eigenstates can appear if $\langle|t_{\mathbf{nm}}|\rangle$ decay as fast as $1/|\mathbf{n}-\mathbf{m}|^d$; (iii) correlations in diagonal disorder $\varepsilon_{\mathbf{n}}$ weaken localization while correlations in off-diagonal disorder $t_{\mathbf{nm}}$ can favour localization.

We now discuss the characteristics and properties of the different models we used, as well as their link with the Anderson model (2). A summary is given in Table 1.

## 2.1 Power-Law Random Banded Matrices (PRBM)

Power-law random banded matrices were first introduced in [48]. They were inspired from earlier random banded matrix ensembles with exponential decay describing the transition from integrability to chaos [49]. The PRBM model is defined by symmetric or Hermitian matrices whose elements are identical independently distributed (i.i.d.) Gaussian random variables with zero mean and variance decreasing as a power law with the distance from the diagonal. The critical PRBM model corresponds to an Anderson model (2) with random long-range hopping whose variance decays as the inverse of the distance between sites.

More precisely, let $\mathcal{N}(\mu, \sigma)$ be a Gaussian distribution of mean $\mu$ and standard deviation $\sigma$. In the following we use the version of PRBM considered in [17, 50], with periodic boundary conditions, where for $N \times N$ matrices diagonal entries $\varepsilon_n$ are i.i.d. with distribution $\mathcal{N}(0, 1)$, and real and imaginary parts of the off-diagonal entries $t_{mn}$ are i.i.d. with distribution $\mathcal{N}\left(0, \sigma_{nm}/\sqrt{2}\right)$,

$$\sigma_{nm}^2 = \left(1 + \frac{\sin^2(\pi|n-m|/N)}{(b\pi/N)^2}\right)^{-1}. \tag{3}$$

In particular we have $\sqrt{\langle|t_{nm}|^2\rangle} = \sigma_{nm}$, which scales as $\sim 1/|n-m|$ for $b \ll |n-m| \ll N$.

The density of states is defined as

$$\rho(E) = \langle \frac{1}{N^d} \sum_{\alpha} \delta(E - \omega_{\alpha}) \rangle \,, \tag{4}$$

which for this model gives

$$\rho_{\text{PRBM}}(E) = \begin{cases} \frac{1}{\sqrt{2\pi}} \exp\left(-\frac{E^2}{2}\right), & b \ll 1, \\ \frac{1}{2b\pi^2}\sqrt{4b\pi - E^2}, & b \gg 1. \end{cases} \tag{5}$$

Eigenvectors are multifractal, and their multifractal dimensions $D_q$, which depend on both $E$ and parameter $b$, can be analytically computed [18,50]. Parameter $b$ makes it possible to explore the whole range of multifractality regime: the weak multifractality regime $D_q \to 1$ is reached for $b \to \infty$ and the strong multifractality regime $D_q \to 0$ is reached for $b \to 0$. All numerical data presented in this work are performed at the center of the band $E = 0$.

## 2.2 Ruijsenaars-Schneider model

Let us consider the following deterministic kicked rotor model [51,52]

$$\hat{H} = \frac{\tau \hat{p}^2}{2} + V(\hat{x}) \sum_n \delta(t - n), \tag{6}$$

with a $2\pi$-periodic sawtooth potential $V(x) = ax$ for $-\pi < x < \pi$, and where $\tau$ is a constant parameter. As a direct consequence of spatial periodicity of $V(x)$, momenta only take quantized values $p_n = 0, \pm1, \pm2, \dots$ (here $\hbar = 1$). Additionally, we consider a truncated basis in $p$ space, with periodic boundary conditions, so that the total number of momenta states $|p_n\rangle$ accessible is $N$. This implies that position basis is also discretized ($x_k$ are separated by intervals $2\pi/N$, with $k$ an integer).

It is well-known that kicked Hamiltonians such as (6) can be mapped onto the Anderson model (2) [53,54]. The $N$ quantized plane waves $|p_n\rangle$ then play the role of lattice site states $|n\rangle$. The mapping is given (for an eigenvector of the Floquet operator with eigenphase $e^{i\omega}$) by

$$\varepsilon_n = \tan\left(\omega/2 - \tau n^2/4\right), \tag{7}$$

$$t_{nm} = -\int_{-\pi}^{\pi} \frac{dx}{2\pi} \tan[V(x)/2] e^{-ix(m-n)}, \tag{8}$$

where the on-site energy $\varepsilon_n$ takes evenly distributed pseudo-random values, provided $\tau$ is sufficiently irrational. As a consequence of the Fourier transform relation in Eq. (8), discontinuity of the sawtooth potential $V(x)$ creates a long-range decay of the couplings $t_{nm} \sim 1/|n-m|$ and actually induces multifractal eigenstates.

The Ruijsenaars-Schneider (RS) model was introduced in the context of classical mechanics [55–57]. Its quantum properties were studied in [58–60]. It is defined (for an arbitrary real parameter $a$) by the Floquet operator of the Hamiltonian (6) (with truncated basis in $p$ space)

$$\hat{U} = e^{-i\varphi_{\hat{p}}} e^{-ia\hat{x}}, \tag{9}$$

where the deterministic kinetic phase has been replaced by random phases $\varphi_{\hat{p}}$ (consequently the on-site energies $\varepsilon_n$ in Eq. (7) are truly uncorrelated), and $x$ is taken modulo $2\pi$ [61].

Importantly, unlike for PRBM, eigenstate properties of the RS matrix ensemble do not depend on their quasi-energy. In particular it has a flat density of states

$$\rho_{\text{RS}}(E) = \frac{1}{2\pi}. \tag{10}$$

Eigenvectors are multifractal; the multifractal dimensions can be derived in certain perturbation regimes, and only depend on the parameter $a$ [62–65]. This parameter $a$ allows us to explore the whole range of multifractality regimes : the weak multifractality regime $D_q \to 1$ is reached for $a \to 1$ and the strong multifractality regime $D_q \to 0$ is reached for $a \to 0$.

### 2.3   3d Random Kicked Rotor (3DKR)

Our three-dimensional (3d) model is the deterministic kicked rotor, defined by the following Hamiltonian [66]

$$\hat{H} = \frac{\tau_x p_x^2}{2} + \frac{\tau_y p_y^2}{2} + \frac{\tau_z p_z^2}{2} + V(\mathbf{q})\sum_n \delta(t-n), \tag{11}$$

where $\tau_i$ are constant parameters and the spatial potential writes $V(\mathbf{q}) = K\mathcal{V}(x)\mathcal{V}(y)\mathcal{V}(z)$, $K$ the kick strength, with

$$\mathcal{V}(x) = \frac{\sqrt{2}}{2}\left(\cos x + \frac{1}{2}\sin 2x\right), \tag{12}$$

so that the system breaks the time-reversal symmetry [45].

As previously stated, the Hamiltonian (11) can be mapped onto the 3d Anderson model (2). For a given eigenstate of the system with eigenphase $e^{i\omega}$, this mapping writes

$$\varepsilon_{\mathbf{n}} = \tan\left(\omega/2 - \tau_x n_x^2/4 - \tau_y n_y^2/4 - \tau_z n_z^2/4\right), \tag{13}$$

$$t_{\mathbf{nm}} = -\iint_{-\pi}^{\pi} \frac{d\mathbf{q}}{(2\pi)^3}\tan[V(\mathbf{q})/2]e^{-i\mathbf{q}\cdot(\mathbf{m}-\mathbf{n})}, \tag{14}$$

where energies $\varepsilon_{\mathbf{n}}$ take pseudo-random values (provided that $(\tau_x, \tau_y, \tau_z)$ are incommensurate numbers), and where hopping terms $t_{\mathbf{nm}}$ decay exponentially fast with distance between sites $|\mathbf{n}-\mathbf{m}|$ [66].

The 3d random Kicked Rotor (3DKR) that we consider in the following corresponds to the Floquet operator of Hamiltonian (11)

$$\hat{U} = e^{-i\phi_{\hat{\mathbf{p}}}}e^{-iV(\hat{\mathbf{q}})}, \tag{15}$$

where deterministic kinetic phases are replaced by uniformly distributed random phases $\phi_{\mathbf{p}}$ (this implies in particular that energies $\varepsilon_{\mathbf{n}}$ in Eq. (13) are uncorrelated).

The 3DKR can be seen as the Floquet counterpart of the usual 3d unitary Anderson Model. In particular it undergoes an Anderson transition monitored by the parameter $K$ (that is related to the hopping intensity). Using techniques inspired by [66, 67], we found that the critical value is $K_c \approx 1.58$ (see Appendix A). However, unlike the 3d Anderson model, this unitary counterpart has a flat density of states

$$\rho_{3DKR}(E) = \frac{1}{2\pi}, \tag{16}$$

and no mobility edge.

Furthermore, we assumed that 3DKR has the same multifractal dimensions as the corresponding unitary 3d Anderson model, because it belongs to the same universality class. The values that were determined in [68] (using the same techniques as in [69, 70]) are $D_1 = 1.912 \pm 0.007$ and $D_2 = 1.165 \pm 0.015$.

## 3   General framework for the study of CFS in critically disordered systems

### 3.1   Eigenstates and time propagator

In the following, we will analytically and numerically address CFS in critical disordered systems within a very general framework, including both Floquet and Hamiltonian cases. The numerical methods are presented in Appendix B.

For the sake of clarity, we use a common notation: $|\phi_\alpha\rangle$ refer to eigenstates (or Floquet modes) with energy (or quasienergy) $\omega_\alpha$. The time propagator of the system then writes

$$\hat{U}(t) = \sum_\alpha e^{-i\omega_\alpha t} |\phi_\alpha\rangle \langle \phi_\alpha| \,, \tag{17}$$

where time will be considered a continuous variable. In particular, we use the following convention and notation for the temporal Fourier transform:

$$f(\omega) = \int_{-\infty}^{\infty} dt \, f(t) e^{i\omega t} \,, \qquad f(t) = \int_{-\infty}^{\infty} \frac{d\omega}{2\pi} f(\omega) e^{-i\omega t} \,. \tag{18}$$

## 3.2 Direct and reciprocal spaces

As illustrated by the models introduced above, in generic critical disordered systems disorder can be present either in position space (e.g. PRBM, Anderson model) or momentum space (e.g. 3DKR, RS). From now on, we refer to the basis where disorder is present (labeled with kets $|\mathbf{n}\rangle$) as the *direct space* and to its Fourier-conjugated basis (labeled with kets $|\mathbf{k}\rangle$) as the *reciprocal space*. This distinction is particularly important because multifractality of eigenstates is a basis-dependent property that only appears in *direct space*, where disorder is present, while CFS is an interference effect taking place in *reciprocal space*.

Importantly, we choose to use standard notations of spatially disordered lattice systems, as in Eq. (2). For a $d$-dimensional system, direct space is spanned by discrete lattice sites states $|\mathbf{n}\rangle = |n_1, \ldots, n_d\rangle$ ($n_i = -N/2 + 1, \ldots N/2$) ($N$ will be considered even). The dimension of the associated Hilbert space is $N^d$. Consequently, the reciprocal space is spanned by a basis $|\mathbf{k}\rangle = |k_1, \ldots, k_d\rangle$ (where $k_i = \pm\frac{\pi}{N}, \pm\frac{3\pi}{N} \cdots \pm \frac{(N-1)\pi}{N}$). We also choose the following convention for the change of basis (see Appendix C for details)

$$\phi_\alpha(\mathbf{k}) = \sum_{\mathbf{n} \in \,]-N/2, N/2]^d} \phi_\alpha(\mathbf{n}) e^{-i\mathbf{k}\cdot\mathbf{n}} \,, \tag{19}$$

$$\phi_\alpha(\mathbf{n}) = \frac{1}{N^d} \sum_{\mathbf{k} \in \,]-\pi, \pi]^d} \phi_\alpha(\mathbf{k}) e^{i\mathbf{k}\cdot\mathbf{n}} \,, \tag{20}$$

so that in the limit $N \to \infty$ the system tends to a infinite-size *discrete* lattice, that is,

$$\phi_\alpha(\mathbf{k}) \xrightarrow[N\to\infty]{} \sum_{n_1=-\infty}^{\infty} \cdots \sum_{n_d=-\infty}^{\infty} \phi_\alpha(\mathbf{n}) e^{-i\mathbf{k}\cdot\mathbf{n}} \,, \tag{21}$$

$$\phi_\alpha(\mathbf{n}) \xrightarrow[N\to\infty]{} \iint_{-\pi}^{\pi} \frac{d^d\mathbf{k}}{(2\pi)^d} \phi_\alpha(\mathbf{k}) e^{i\mathbf{k}\cdot\mathbf{n}} \,. \tag{22}$$

We insist that for 3DKR and RS models, *direct space* is the momentum space. For instance for the RS model the basis $|\mathbf{n}\rangle$ corresponds to plane waves with discrete momenta $p = n\hbar$ (with $\hbar = 1$) because of spatial $2\pi$-periodicity of kicked Hamiltonians. Consequently, the *reciprocal space* corresponds to position space, so that $|\mathbf{k}\rangle$ corresponds to discrete positions $x_k = \pm\frac{\pi}{N}, \pm\frac{3\pi}{N} \cdots \pm \frac{(N-1)\pi}{N}$. Spatial discretization comes from the imposed periodic boundary conditions in the truncated momentum basis, so that the linear system size in direct space is $N$.

## 3.3 Form factor and level compressibility

Previous studies [39–46] found that CFS dynamics could be related to the form factor. We will show that it is the same in critical disordered systems. We recall some definitions that will be useful in forthcoming calculations.

### 3.3.1 Form factor

The form factor is the Fourier transform of the two-point energy correlator; it is usually defined as

$$K_N(t) = \frac{1}{N^d} \left\langle \sum_{\alpha,\beta} e^{-i\omega_{\alpha\beta}t} \right\rangle , \tag{23}$$

with $\omega_{\alpha,\beta} = \omega_\beta - \omega_\alpha$. It can be rewritten as

$$K_N(t) = \int dE\, \rho(E) K_N(t; E), \tag{24}$$

with

$$K_N(t; E) = \frac{1}{N^d \rho(E)} \left\langle \sum_{\alpha\beta} e^{-i\omega_{\alpha\beta}t} \delta\left(E - \frac{\omega_\alpha + \omega_\beta}{2}\right) \right\rangle . \tag{25}$$

The component $K_N(t; E)$ of the form factor can be interpreted as coming from contributions of all interfering pairs of states whose average energy is $E$. In order to lighten forthcoming calculations, we introduce the following implicit notation

$$\left\langle \sum_{\alpha,\beta} \dots \right\rangle_E \equiv \left\langle \frac{1}{\rho(E)} \sum_{\alpha,\beta} \delta\left(E - \frac{\omega_\alpha + \omega_\beta}{2}\right) \dots \right\rangle , \tag{26}$$

$$\langle f(\omega_\alpha) \rangle_E \equiv \left\langle \frac{1}{\rho(E)} \sum_\alpha \delta(E - \omega_\alpha) f(\omega_\alpha) \right\rangle , \tag{27}$$

so that $K_N(t; E)$ writes

$$K_N(t; E) = \frac{1}{N^d} \left\langle \sum_{\alpha,\beta} e^{-i\omega_{\alpha\beta}t} \right\rangle_E . \tag{28}$$

### 3.3.2 Compressibility and link to multifractal dimensions

The level compressibility $\chi$ is defined as

$$\chi = \lim_{t/N^d \to 0} K_N(t; E). \tag{29}$$

It is a measure of long-range correlations in the spectrum. It estimates how much the variance of the number of states in a given energy window scales with the size of the window. For usual random matrices (GOE, GUE...) $\chi = 0$, while for Poisson statistics $\chi = 1$.

For critical systems that have intermediate statistics, the level compressibility lies in between $0 < \chi < 1$ [27]. It was proposed that $\chi$ could actually be related to multifractal dimension $D_2$ via $\chi = 1 - D_2/2d$ [27, 71], but it was later observed that this relation fails in the weak multifractal regime. Another relation was then conjectured [60], relating $\chi$ to the information dimension $D_1$

$$\chi = 1 - \frac{D_1}{d}, \tag{30}$$

and has since been verified in many different systems [63, 72–74] (see also Appendix B).

The information dimension $D_1$ appearing in Eq. (30) is defined trough the asymptotic expansion of Eq. (1) in the limit $q \to 1$

$$\frac{\langle \sum_{\mathbf{n},\alpha} \delta(E - \omega_\alpha) |\phi_\alpha(\mathbf{n})|^2 \ln |\phi_\alpha(\mathbf{n})|^2 \rangle}{\langle \sum_\alpha \delta(E - \omega_\alpha) \rangle} \sim D_1 \ln N , \tag{31}$$

and can be seen as the Shannon entropy of eigenstates $|\phi_\alpha(\mathbf{n})|^2$.

## 3.4 Energy decomposition and contrast definition

CFS is an interference effect that appears when the system is initially prepared in a state localized in reciprocal space, $|\psi(t=0)\rangle = \frac{1}{N^{d/2}}|\mathbf{k}_0\rangle$ (our Fourier transform and normalization conventions are listed in Appendix C). The observable of interest is the disorder averaged scattering probability in direction $\mathbf{k}$, defined as $n(\mathbf{k},t) = \frac{1}{N^d}\langle|\langle\mathbf{k}|\hat{U}(t)|\mathbf{k}_0\rangle|^2\rangle$. Using (17), it can be expanded over eigenstates as

$$n(\mathbf{k},t) = \frac{1}{N^d}\langle\sum_{\alpha,\beta}e^{-i\omega_{\alpha\beta}t}\phi_\alpha(\mathbf{k})\phi_\alpha^\star(\mathbf{k}_0)\phi_\beta(\mathbf{k}_0)\phi_\beta^\star(\mathbf{k})\rangle. \tag{32}$$

### 3.4.1 Energy decomposition

As previously stated, multifractal properties of eigenstates may depend on their energy. Following the lines of [44], we rewrite the contrast in the following way

$$n(\mathbf{k},t) = \int dE\,\rho(E)n(\mathbf{k},t;E), \tag{33}$$

where $n(\mathbf{k},t;E)$ is the contribution of all interfering pairs of states whose average energy is $E$ and is given by (see Eqs. (26)–(27))

$$n(\mathbf{k},t;E) = \frac{1}{N^d}\langle\sum_{\alpha,\beta}e^{-i\omega_{\alpha\beta}t}\phi_\alpha(\mathbf{k})\phi_\alpha^\star(\mathbf{k}_0)\phi_\beta(\mathbf{k}_0)\phi_\beta^\star(\mathbf{k})\rangle_E. \tag{34}$$

### 3.4.2 Classical incoherent background

Coherent scattering effects (such as CFS and CBS) build on top of a classical incoherent diffusive background. This classical incoherent contribution can be described by introducing the disorder-averaged spectral function

$$A(\mathbf{k};E) = \frac{1}{N^d}\langle\sum_\alpha|\phi_\alpha(\mathbf{k})|^2\delta(E-\omega_\alpha)\rangle. \tag{35}$$

Using the normalization condition Eq. (C.14), $A(\mathbf{k}_0;E)$ can be interpreted as the probability that the system has energy $E$ when initialized in the plane wave state $|\mathbf{k}_0\rangle$. By the same token, Eq. (C.13) shows that $A(\mathbf{k},E)/\rho(E)$ can be interpreted as the distribution in reciprocal space associated with the system residing on the energy-shell $E$ (ergodicity). Taking the product of these two probabilities and using Eq. (33), we find that the classical incoherent contribution reads:

$$n_{\text{class}}(\mathbf{k};E) = \frac{A(\mathbf{k},E)}{\rho(E)}\frac{A(\mathbf{k}_0,E)}{\rho(E)}. \tag{36}$$

This result has been derived and numerically checked in [41,42] in the case of random potentials in 1 or 2 dimensions (note that in these works one of the factors $\rho(E)$ in the denominator was absorbed in the definition of the spectral function at energy $E$).

For usual disordered systems such as the Anderson model, the spectral function $A(\mathbf{k};E)$ depends on $\mathbf{k}$ with a width related to the inverse scattering mean free path $1/\ell_s$ [42]. However, for kicked systems such as models (9) and (15), one can show that $A(\mathbf{k};E) = \rho(E)$ [45]. The essence of the argument is that the Fourier transform of (35) in direct space $\mathbf{n}$ and time $A(\mathbf{n};t)$ is given by the matrix elements of $\hat{U}^t$ averaged over disorder:

$$\langle\langle\mathbf{m}|\hat{U}^t|\mathbf{n}\rangle\rangle = \delta_{\mathbf{m},\mathbf{n}}\delta_{t,0}. \tag{37}$$

This result is a consequence of the uniform distribution of the random phases over $[0, 2\pi]$. The equality $A(\mathbf{k}; E) = \rho(E)$ can be seen as the limit $\ell_s \to 0$, that is, when $\ell_s$ becomes less than the lattice spacing [41]. Notably, we found that the relation $A(k; E) = \rho(E)$ also holds in the case of PRBM, where the inverse scattering mean free path is less clearly defined; this is illustrated in Fig. 8 of Appendix B. In fact, for PRBM the relation is a consequence of the independence of the matrix elements, as we demonstrate analytically in Appendix D.

This property that the spectral function reduces to the density of states can be understood as a consequence of a "diagonal approximation" central to our work. Starting from (35) and expanding $A(\mathbf{k}; E)$ in direct space, we have

$$A(\mathbf{k}; E) = \frac{1}{N^d} \sum_{\mathbf{n}, \mathbf{m}} \langle \sum_\alpha \phi_\alpha(\mathbf{n}) \phi_\alpha^\star(\mathbf{m}) \delta(E - \omega_\alpha) \rangle \, e^{i\mathbf{k} \cdot (\mathbf{n} - \mathbf{m})} \,. \tag{38}$$

The case where disorder average washes out the off-diagonal terms $\mathbf{n} \neq \mathbf{m}$ is usually referred to as "diagonal approximation". Under that approximation we have

$$A(\mathbf{k}; E) \approx \frac{1}{N^d} \sum_{\mathbf{n}} \langle \sum_\alpha |\phi_\alpha(\mathbf{n})|^2 \delta(E - \omega_\alpha) \rangle = \rho(E) \,. \tag{39}$$

The identity $A(k; E) = \rho(E)$ can thus be seen as resulting from the absence of correlations between norm and phase of the eigenstates in direct space, so that only terms where phase factors cancel (i.e. diagonal elements) do survive the disorder average. This is corroborated by the direct numerical computation of these correlations for the RS model (see Appendix E), as well as by the analytical derivation of Appendix D in the PRBM case. We thus think that the diagonal approximation we use in this article should hold in many critical systems, as long as there is no correlation in disorder that might induce correlations between norm and phase in direct space.

The classical contribution Eq. (36) then simply reduces to a $\mathbf{k}$-independent and $E$-independent flat background $n_{\text{class}}(\mathbf{k}; E) = 1$.

### 3.4.3 Contrast

The CFS and CBS peaks emerge from this classical background. Following the lines of [44] we introduce the CFS contrast $\Lambda_N(\mathbf{k}, t; E)$ as the interference pattern relative to the classical background, at a given energy. In the diagonal approximation discussed above, it simply reads

$$\Lambda_N(\mathbf{k}, t; E) = n(\mathbf{k}, t; E) - 1 \,. \tag{40}$$

## 4 Universal predictions for CFS dynamics

In this Section we explain the main hypotheses of our approach, and we derive a simple expression for the CFS contrast. We then discuss the existence of two distinct dynamical regimes, one corresponding to large time limit of finite-size systems, the other one to infinite-size systems. We describe the CFS contrast in these two regimes.

### 4.1 General predictions

#### 4.1.1 Extended diagonal approximation

First, we take the temporal Fourier transform (18) of the CFS contrast given by (34) and (40), and expand it in direct space. This gives

$$\Lambda_N(\mathbf{k}, \omega; E) = \frac{2\pi}{N^d} \sum_{\substack{\mathbf{n}_1, \mathbf{n}_2 \\ \mathbf{n}_3, \mathbf{n}_4}} C(\omega; E) e^{i\mathbf{k} \cdot (\mathbf{n}_1 - \mathbf{n}_4) - i\mathbf{k}_0 \cdot (\mathbf{n}_2 - \mathbf{n}_3)} - 2\pi \delta(\omega) \,, \tag{41}$$

with

$$C(\omega; E) = \langle \sum_{\alpha,\beta} \delta(\omega - \omega_{\alpha\beta}) \phi_\alpha(\mathbf{n}_1) \phi_\alpha^\star(\mathbf{n}_2) \phi_\beta(\mathbf{n}_3) \phi_\beta^\star(\mathbf{n}_4) \rangle_E .$$ (42)

Following the idea of the "diagonal approximation" used to derive Eq. (39), we claim that correlation functions $C(\omega; E)$ should generically vanish (or become negligible) upon disorder average unless they are of the two following kinds: (i) tuples such as $\mathbf{n}_1 = \mathbf{n}_2$ and $\mathbf{n}_3 = \mathbf{n}_4$, that give a real positive contribution, and (ii) tuples such as $\mathbf{n}_1 = \mathbf{n}_4 \equiv \mathbf{n}$ and $\mathbf{n}_2 = \mathbf{n}_3 \equiv \mathbf{m}$, whose temporal Fourier transform is the average transfer probability (at a given energy $E$) between $|\mathbf{n}\rangle$ and $|\mathbf{m}\rangle$ in *direct* space, namely

$$\langle | \langle \mathbf{n}|\hat{U}(t)|\mathbf{m}\rangle |^2 \rangle_E = \langle \sum_{\alpha,\beta} e^{-i\omega_{\alpha\beta}t} \phi_\alpha(\mathbf{n}) \phi_\alpha^\star(\mathbf{m}) \phi_\beta(\mathbf{m}) \phi_\beta^\star(\mathbf{n}) \rangle_E .$$ (43)

### 4.1.2 Compact approximate expression for the contrast

Keeping only these non-vanishing contributions (and taking care of double count of the tuple $\mathbf{n}_1 = \mathbf{n}_2 = \mathbf{n}_3 = \mathbf{n}_4$), the CFS contrast can be approximated by

$$\Lambda_N(\mathbf{k}, \omega; E) = \Lambda^{(1)} + \Lambda^{(2)} - 2\pi\delta(\omega),$$ (44)

where the first term corresponds to the contribution $\mathbf{n}_1 = \mathbf{n}_2$ and $\mathbf{n}_3 = \mathbf{n}_4$,

$$\Lambda^{(1)} = \frac{2\pi}{N^d} \sum_{\mathbf{n}\neq\mathbf{m}} \langle \sum_{\alpha,\beta} \delta(\omega - \omega_{\alpha\beta})|\phi_\alpha(\mathbf{n})|^2 |\phi_\beta(\mathbf{m})|^2 \rangle_E \, e^{i(\mathbf{k}-\mathbf{k}_0)\cdot(\mathbf{n}-\mathbf{m})},$$ (45)

and the second term comes from the contribution $\mathbf{n}_1 = \mathbf{n}_4$ and $\mathbf{n}_2 = \mathbf{n}_3$,

$$\Lambda^{(2)} = \frac{2\pi}{N^d} \left\langle \sum_{\alpha,\beta} \delta(\omega - \omega_{\alpha\beta})\delta_{\alpha\beta} \right\rangle_E = 2\pi\delta(\omega).$$ (46)

In (46), the Kronecker delta $\delta_{\alpha\beta}$ appears because of eigenstate orthonormalization, and simplifications arises from Eq. (26), using the definition (4) of the density of states. The second term $\Lambda^{(2)}$ thus exactly compensates the Dirac delta in (44). The CFS contrast reduces to $\Lambda^{(1)}$, and is finally given by the following compact expression

$$\Lambda_N(\mathbf{k}, \omega; E) = 2\pi \sum_{\mathbf{n}\neq 0} \langle \sum_{\alpha,\beta} \delta(\omega - \omega_{\alpha\beta})|\phi_\alpha(\mathbf{n}_0)|^2 |\phi_\beta(\mathbf{n}_0 + \mathbf{n})|^2 \rangle_{E,\mathbf{n}_0} \, e^{-i\mathbf{n}\cdot(\mathbf{k}-\mathbf{k}_0)},$$ (47)

or equivalently

$$\Lambda_N(\mathbf{k}, t; E) = \sum_{\mathbf{n}\neq 0} \langle \sum_{\alpha,\beta} e^{-i\omega_{\alpha\beta}t}|\phi_\alpha(\mathbf{n}_0)|^2 |\phi_\beta(\mathbf{n}_0 + \mathbf{n})|^2 \rangle_{E,\mathbf{n}_0} \, e^{-i\mathbf{n}\cdot(\mathbf{k}-\mathbf{k}_0)},$$ (48)

where the disorder average $\langle \ldots \rangle_{\mathbf{n}_0}$ additionally runs over different sites $\mathbf{n}_0$.

At the peak $\mathbf{k} = \mathbf{k}_0$, the expression for the CFS contrast further simplifies. Adding and subtracting the contribution $\mathbf{n} = \mathbf{0}$ to the sum in (48) and using normalization of wavefunctions, we get the expression

$$\Lambda_N(\mathbf{k}_0, t; E) = K_N(t; E) - \langle | \langle \mathbf{n}_0|\hat{U}(t)|\mathbf{n}_0\rangle |^2 \rangle_{E,\mathbf{n}_0},$$ (49)

where the first term is the form factor, given by Eq. (28), and second term is the return probability in direct space at energy $E$, see Eq. (43).

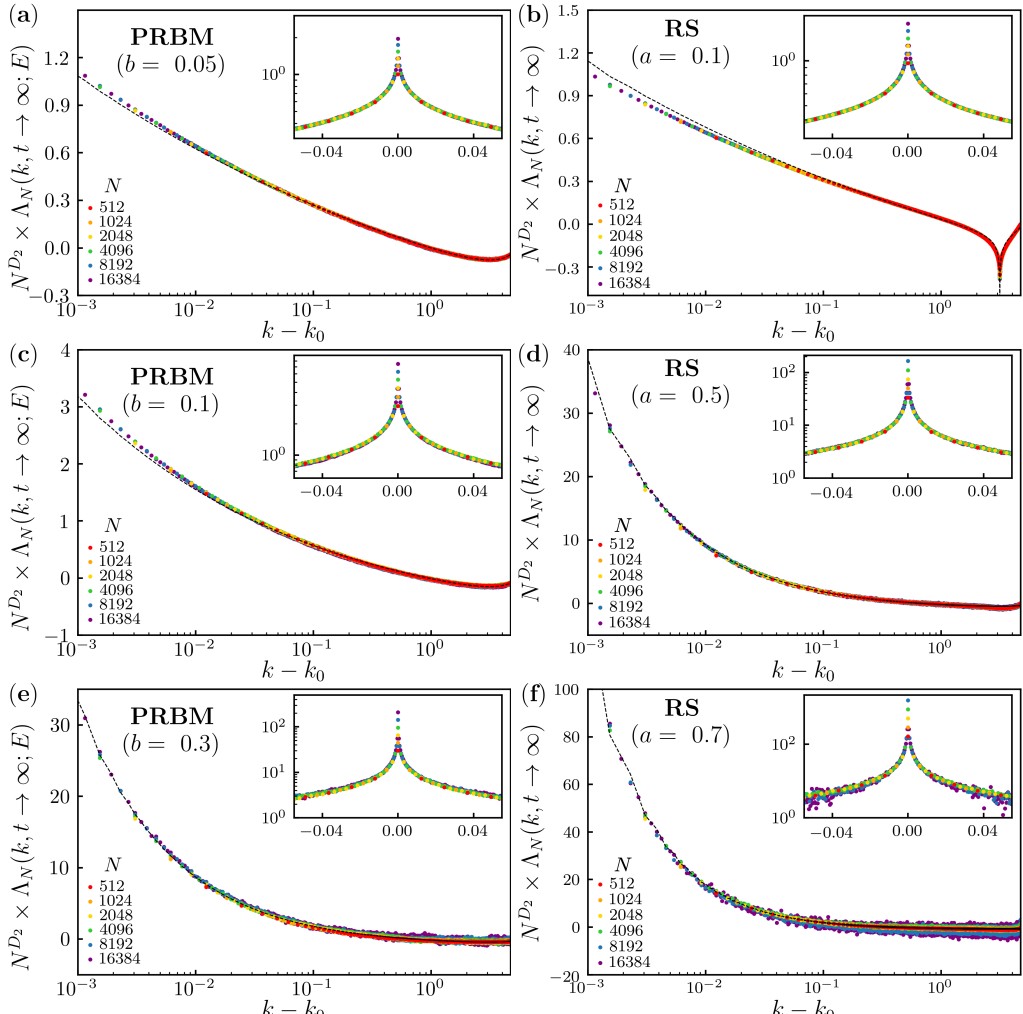

Figure 2: Rescaled CFS contrast for PRBM at $E = 0$ (a,c,e) and for RS averaged over $E$, see Eq. (B.14) (b,d,f) in the limit $t \gg \tau_H$ for different system sizes $N$ (see Appendix B for numerical procedure). Insets are a zoom around $k = k_0$. The dashed line correspond to analytical prediction Eq. (54), with a height fitted far from $k = k_0$ (in panel b, the dashed line corresponds to the symmetrized prediction Eq. (101), where the two parameters $A$ and $B$ have been independently adjusted, which accounts for the anti-peak (see Sec. 5.4)). The value of $D_2$ used in Eq. (54) and in the $y$ axis is obtained from scaling of the moments (1) in direct space.

### 4.1.3 Relevant time scale

It has been shown (see e.g. [42]) that the relevant time scale for the CFS dynamics is given by the Heisenberg time $\tau_H = 2\pi/\Delta$, where $\Delta$ is the mean level spacing. More precisely, the mean level spacing corresponds to the spacing in the confining volume, which is associated to the localization volume in the presence of localization, or to the system volume if the system is delocalized. In the context of critically disordered media, wavefunctions are delocalized (but nonergodic); the mean level spacing is $\Delta = 1/(N^d \rho(E))$, which depends on the system size, and thus

$$\tau_H = 2\pi N^d \rho(E). \tag{50}$$

This defines two distinct regimes for the CFS, with specific properties, that we shall explore in turn in the next two subsections: (i) when $t \ll \tau_H$, CFS originates from the nonergodicity

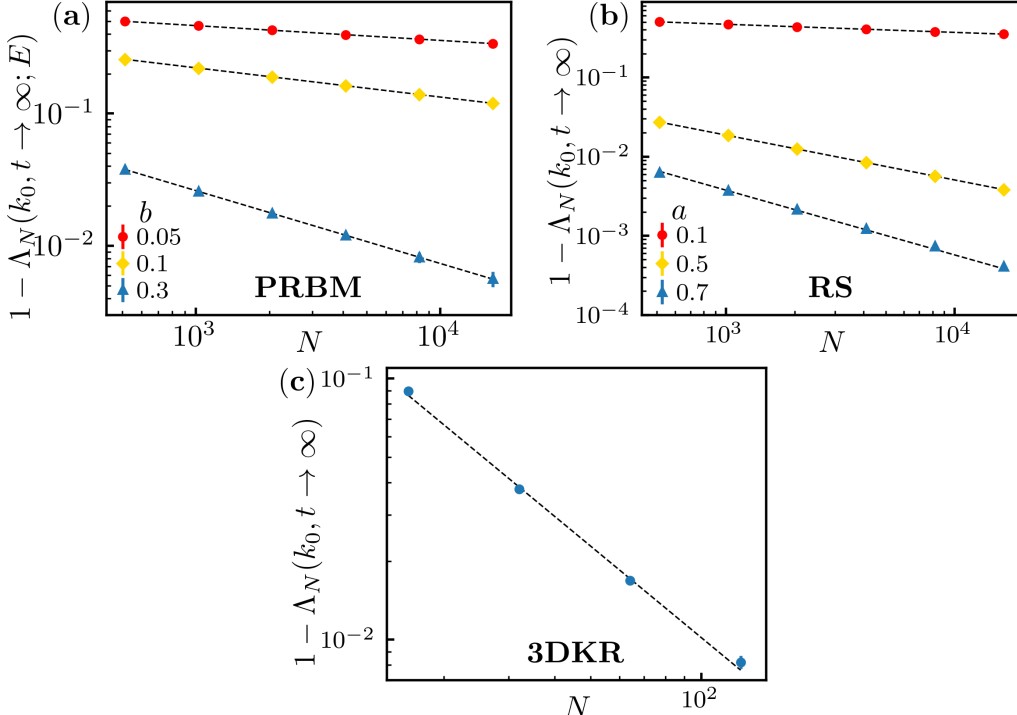

Figure 3: CFS contrast peak in the long-time limit ($t \gg \tau_H$) and its scaling (56) with system size $N$. (a) PRBM model with different $b$ and at $E = 0$. (b) RS model averaged over $E$ with different $a$. (c) 3DKR model with $K = 1.58$. Symbols are numerical data for different system sizes. Dashed black lines are Eq. (56), i.e. a single parameter fit $y = \alpha N^{-D_2}$ with $\alpha$ the fit parameter and $D_2$ independently determined from scaling of the moments (1) in direct space (for PRBM and RS) or taken from [68] (for 3DKR). See Appendix B for numerical procedure.

of the eigenstates ; (ii) when $t \gg \tau_H$, CFS is caused by boundaries of the system. Regime (i) is relevant in the limit of infinite size, which corresponds to the regime numerically explored in [44] in the 3d Anderson model. There it was found that at the AT the height of the CFS peak reaches a stationary value, conjectured to be the compressibility $\chi = 1 - D_1/d$. Regime (ii) corresponds to the long-time limit of a finite-size system.

In the finite-size case, waves travel many times across the entire system until they resolve the discreteness of energy levels. The shape and height of the CFS peak then explicitly depend on system size $N$ (see Section 4.2). When $N$ goes to infinity, the CFS still manifests itself at small times and is due to nonergodicity of eigenstates (see Section 4.3). This is to be contrasted with the localized regime of the Anderson transition, where the behavior differs depending on whether the localization length is smaller or larger than the system size.

## 4.2 Long-time limit

### 4.2.1 CFS peak shape

We now discuss the long-time limit in finite-size systems i.e. the regime $t \gg \tau_H$, $t \to \infty$ with fixed system size $N$. The contrast defined by (34) and (40) is then only determined by diagonal terms $\omega_{\alpha\beta} = 0$ (which are the only ones that survive the long-time limit), so that the expression of the contrast is given by

$$\Lambda_N(\mathbf{k}, t \to \infty; E) = \frac{1}{N^d} \langle |\phi_\alpha(\mathbf{k})|^2 |\phi_\alpha(\mathbf{k}_0)|^2 \rangle_E - 1. \tag{51}$$

On the other hand, using the same argument, the approximate expression (48) can be rewritten as

$$\Lambda_N(\mathbf{k}, t \to \infty; E) = \sum_{\mathbf{n} \neq 0} \langle |\phi_\alpha(\mathbf{n}_0)|^2 |\phi_\alpha(\mathbf{n}_0 + \mathbf{n})|^2 \rangle_{E,\mathbf{n}_0} \, e^{-i\mathbf{n}\cdot(\mathbf{k}-\mathbf{k}_0)}. \tag{52}$$

This expression can be seen as the spatial Fourier transform of the two-point correlator in direct space. For a function which is multifractal in direct space the correlator has the asymptotic behavior [18]

$$N^d \langle |\phi_\alpha(\mathbf{n}_0)|^2 |\phi_\alpha(\mathbf{n}_0 + \mathbf{n})|^2 \rangle_{E,\mathbf{n}_0} \sim \left| \frac{N}{\mathbf{n}} \right|^{d-D_2}. \tag{53}$$

It implies that the CFS contrast shape in the long-time limit can be approximated (up to a prefactor) by

$$\frac{\Lambda_N(\mathbf{k}, t \to \infty; E)}{N^{-D_2}} \sim \sum_{|\mathbf{n}| \geq 1} \frac{\cos[\mathbf{n} \cdot (\mathbf{k} - \mathbf{k}_0)]}{|\mathbf{n}|^{d-D_2}}. \tag{54}$$

The right-hand term only depends on $\mathbf{k}$ and $D_2$, and becomes $N$-independent for $N$ sufficiently large. The behavior (54) is confirmed by the numerical simulations displayed in Fig. 2, which show that all the curves $N^{D_2}\Lambda(\mathbf{k}, t)$ collapse onto the predicted expression.

We note however a strong discrepancy when $\mathbf{k} \to \mathbf{k}_0$ in the insets of Fig. 2. This comes from the existence of a high spatial cut-off for the scaling law (53), roughly given by the system size $N$. As a consequence, (54) fails to describe the CFS distribution on a scale smaller than $|\delta \mathbf{k}| \sim 2\pi/N$.

In the specific case of RS model when $a \to 0$, we also note the appearance of an anti-CBS peak (see Figs. 2b and 5b) that comes from a nontrivial asymptotic symmetry of the system and is not relevant in the general case (it is not present in PRBM and 3DKR). We give a more detailed account of this specificity in Sec. 5.4.

### 4.2.2 CFS height

Although (54) fails to describe the CFS distribution at $\mathbf{k} = \mathbf{k}_0$, it is actually possible to circumvent this limitation starting back from (52) and rewriting it for $\mathbf{k} = \mathbf{k}_0$ as

$$\Lambda_N(\mathbf{k}_0, t \to \infty; E) = \sum_{\mathbf{n}} \langle |\phi_\alpha(\mathbf{n}_0)|^2 |\phi_\alpha(\mathbf{n}_0 + \mathbf{n})|^2 \rangle_{E,\mathbf{n}_0} - \langle |\phi_\alpha(\mathbf{n}_0)|^4 \rangle_{E,\mathbf{n}_0}. \tag{55}$$

The first term is actually equal to 1 from eigenstate normalization. The second term is nothing but the inverse participation ratio (up to a factor $N$). It gives the following scaling law

$$1 - \Lambda_N(\mathbf{k}_0, t \to \infty; E) \sim N^{-D_2}. \tag{56}$$

Note that this result could alternatively by obtained from Eq. (49) in the limit $t \to \infty$. Indeed at large $t$ the form factor goes to 1, while the return probability behaves as $N^{-D_2}$ [27].

The scaling dependence (56) is illustrated in Fig. 3 for the three models investigated here. This shows that the long-time behavior of the CFS peak allows us to extract the multifractal dimension $D_2$.

### 4.3 Limit of infinite system size

We now discuss the CFS contrast dynamics in the limit $N \to \infty$, at fixed time $t \ll \tau_H$. In this regime, as we will see below, CFS arises from the nonergodicity of the eigenstates, and it no longer depends on $N$.

### 4.3.1 Dynamics of the CFS at $\mathbf{k} = \mathbf{k}_0$

At the peak the contrast is given by Eq. (49). In the limit $t \ll \tau_H$, the spectral form factor goes to the compressibility $\chi$, while the return probability follows a temporal power law decay related to the multifractal dimension $D_2$ [27,75]

$$\langle |\langle \mathbf{n}_0|\hat{U}(t)|\mathbf{n}_0\rangle|^2\rangle_{E,\mathbf{n}_0} \sim t^{-D_2/d} \, . \tag{57}$$

The height of the CFS peak is then finally given by

$$\Lambda_{N\to\infty}(\mathbf{k}_0, t; E) = \chi - \alpha t^{-D_2/d} \, , \tag{58}$$

where $\alpha$ is a constant that may depend on $E$ (but not on $N$ and $t$). If we assume that the relation (30) between compressibility and information dimension holds, then measuring the time dependence of the peak height at small times allows us to access $D_1$. This is illustrated in Fig. 4 (left panels), where the contrast is plotted as a function of time for the three models discussed here. A proper rescaling of the curves allows to extract $D_1$ as the constant small-time behavior of the CFS contrast.

### 4.3.2 Dynamics of the CFS contrast shape

We now discuss more generally the dynamics of the CFS contrast shape. To do so, we use the fact that the two following correlation functions behave in the same way

$$\langle \sum_{\alpha,\beta} \delta(\omega - \omega_{\alpha\beta})\phi_\alpha(\mathbf{n})\phi_\alpha^*(\mathbf{m})\phi_\beta(\mathbf{m})\phi_\beta^*(\mathbf{n})\rangle_E \sim \gamma \langle \sum_{\alpha,\beta} \delta(\omega - \omega_{\alpha\beta})|\phi_\alpha(\mathbf{n})|^2|\phi_\beta(\mathbf{m})|^2\rangle_E \, , \tag{59}$$

with $\gamma$ some constant (see e.g. Eq. 2.32 of [18]). As a consequence, the CFS contrast (48) can be rewritten as

$$\Lambda_N(\mathbf{k}, t; E) = \gamma \sum_{\mathbf{n}} \langle |\langle \mathbf{n}_0|\hat{U}(t)|\mathbf{n}_0 + \mathbf{n}\rangle|^2\rangle_{E,\mathbf{n}_0} e^{-i\mathbf{n}\cdot(\mathbf{k}-\mathbf{k}_0)} - \langle |\langle \mathbf{n}_0|\hat{U}(t)|\mathbf{n}_0\rangle|^2\rangle_{E,\mathbf{n}_0} \, . \tag{60}$$

In the case where $\mathbf{k} = \mathbf{k}_0$ it is easy to check that (60) reduces to

$$\Lambda_N(\mathbf{k}_0, t; E) = \gamma - \langle |\langle \mathbf{n}_0|\hat{U}(t)|\mathbf{n}_0\rangle|^2\rangle_{E,\mathbf{n}_0} \, . \tag{61}$$

This expression coincides with (49) at small $t$ provided $\gamma = \chi$, since the form factor goes to $\chi$ for $t \to 0$. Again, the second term in the above expression is the return probability. The first term in (60) is the spatial Fourier transform of the propagator between two sites in direct space. This quantity is well-known and has been studied in the past, as it plays an important role in the study of the anomalous diffusion in direct space at the transition [16,28,76,77]. Provided $k < 1/l_s$ (with $l_s$ the mean free path, $l_s \sim 1$ in our models) it is a function $f(q)$ of $q = |\mathbf{k}-\mathbf{k}_0|t^{1/d}$ only, that goes to a constant at small argument. In our case, in view of (61) that constant is equal to $\chi$, and thus

$$f(q) = \begin{cases} \chi \, , & q \ll 1 \, , \\ q^{-D_2} \, , & q \gg 1 \, . \end{cases} \tag{62}$$

The CFS contrast (60) finally writes

$$\Lambda_{N\to\infty}(\mathbf{k}, t; E) = f(|\mathbf{k}-\mathbf{k}_0|t^{1/d}) - \alpha t^{-D_2} \, , \tag{63}$$

where $\alpha$ is the same constant as in Eq. (58).

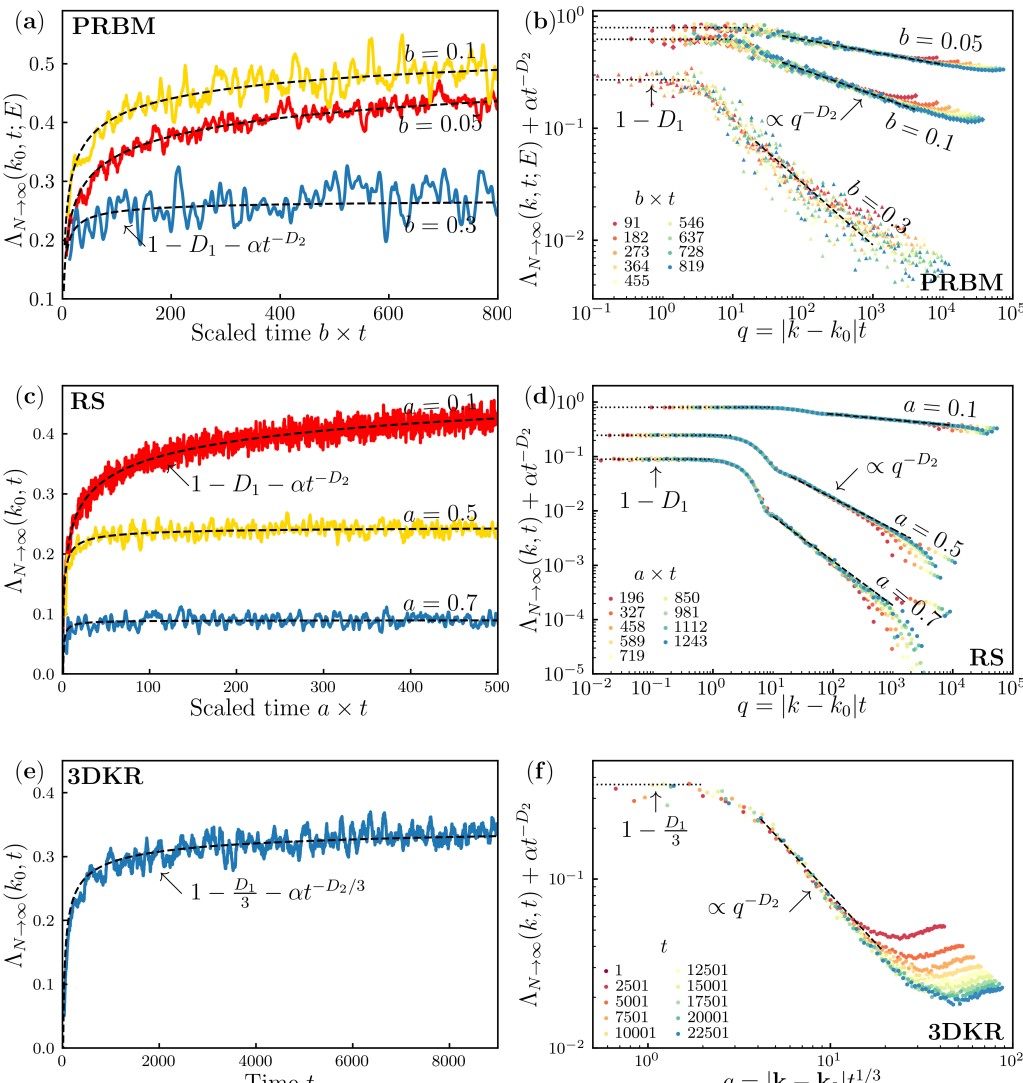

Figure 4: Dynamics of the CFS contrast in the infinite system size limit ($t \ll \tau_H$). (a,b) PRBM model with different $b$, system size $N = 16384$, number of disorder realizations $n_d = 1125$. (c,d) RS model with different $a$, system size $N = 131072$, number of disorder realizations $n_d = 3600$. (e,f) 3DKR model with $K = 1.58$. See Appendix B for various numerical details. (a,c,e) Dynamics of the CFS peak height at $\mathbf{k} = \mathbf{k}_0$. Solid lines are numerical data, smoothed over a range $\Delta t$ for clarity ($\Delta t = 11$ for RS, $\Delta t \sim 10/b$ for PRBM and $\Delta t = 74$ for 3DKR). Dashed black lines are theoretical predictions Eq. (58), i.e. single parameter fit $y = 1 - D_1/d - \alpha t^{-D_2/d}$, with $\alpha$ the fit parameter and $D_1$ and $D_2$ either independently determined from scaling of the moments in direct space (PRBM and RS) or taken from [68] (3DKR). (b,d,f) Dynamics of the CFS peak shape. Symbols are numerical data at different times ($t \in [91/b, 819/b]$ for PRBM, $t \in [196/a, 1243/a]$ for RS, $t \in [1, 22500]$ for 3DKR model). For PRBM and RS models, data are averaged in boxes of $q$ with logarithmically increasing size. For 3DKR, data are averaged over each spherical shell at radius $|\mathbf{k} - \mathbf{k}_0|$. Values of $\alpha$ used to plot the $y$-axis are extracted from the fits presented in (a,c,e). Dashed black lines are a single parameter fit $y = c q^{-D_2}$ (see Eqs. (62) and (63)), with $D_2$ independently determined or taken from literature. Dotted black line is $y = 1 - D_1$, with $D_1$ independently determined or taken from literature.

In Fig. 4, we test these theoretical predictions by comparing them to the numerical data of the three models considered. The left panels represent the temporal dynamics of the CFS contrast at $\mathbf{k}_0$. We clearly observe the convergence towards the compressibility $\chi = 1 - D_1/d$ as time increases, the finite-time effects being controlled by $D_2$, whatever the model and the more or less strong multifractality considered. This confirms Eqs. (62) and (63) for $\mathbf{k} = \mathbf{k}_0$. In the right panels, we represent the spatial dependence of the CFS peak at different times. It is clearly observed that the curves at different times collapse on each other when they are represented as a function of $q$, which confirms the scaling law Eq. (63). Also, the shape of the scaling function $f$ is in perfect agreement with Eq. (62).

## 5 Perturbation theory in the long-time limit and in the strong multifractal regime

### 5.1 Perturbation theory

In this Section we use perturbation theory to derive analytic expressions for the contrast at infinite time in the strong multifractality regime ($D_q \to 0$) of PRBM and RS models (respectively $b \to 0$ and $a \to 0$).

First, we recall that in the long-time limit $t \gg \tau_H$ the CFS contrast Eq. (51) writes

$$\Lambda_N(k, t \to \infty; E) = \frac{\mathcal{I}(E)}{\rho(E)} - 1, \tag{64}$$

with

$$\mathcal{I}(E) = \left\langle \frac{1}{N} \sum_\alpha |\phi_\alpha(k)|^2 |\phi_\alpha(k_0)|^2 \delta(E - \omega_\alpha) \right\rangle. \tag{65}$$

In the following we will find a perturbative expansion of this quantity $\mathcal{I}(E)$ as

$$\mathcal{I}(E) = \mathcal{I}^{(0)}(E) + \mathcal{I}^{(1)}(E) + \dots. \tag{66}$$

To do so, we use a perturbative approach based on the Levitov renormalization-group technique [78]. The idea is that in the strong multifractality regime, the Hamiltonian or Floquet operator $\hat{M}$ is almost diagonal in direct space and the off-diagonal entries $M_{nm} = \langle n|\hat{M}|m \rangle$ can be treated as a perturbation.

At order zero, the operator is diagonal in direct space with eigenvectors given by the canonical basis vectors $|n\rangle$ with energy $E_n = M_{nn}$. It gives

$$\mathcal{I}^{(0)}(E) = \left\langle \sum_n \frac{1}{N} |\langle k|n\rangle|^2 |\langle k_0|n\rangle|^2 \delta(E - E_n) \right\rangle, \tag{67}$$

where the average runs over different disorder realisations of the diagonal entries $M_{nn}$. Using $|\langle k|n\rangle|^2 = 1$ (see Appendix C), we directly get $\mathcal{I}^{(0)}(E) = \rho(E)$: at order 0 the CFS contrast vanishes.

At next order, the main contribution now originates from resonant interactions between pairs of unperturbed states ($|m\rangle, |n\rangle$). They occur if $|H_{mm} - H_{nn}|$ is of the order of $|H_{mn}|$. The corresponding $2 \times 2$ submatrices have two eigenvectors $\left|\phi_{mn}^\mu\right\rangle$ labelled by $\mu = \pm 1$, with energy $E_{mn}^\mu$. The corresponding contribution writes

$$\mathcal{I}^{(1)} = \left\langle \frac{1}{N} \sum_{m<n} \sum_{\mu=\pm} \left|\langle k|\phi_{mn}^\mu\rangle\right|^2 \left|\langle k_0|\phi_{mn}^\mu\rangle\right|^2 \delta(E - E_{mn}^\mu) \right\rangle, \tag{68}$$

where different realizations of random entries $M_{nm}$ will lead to different pairs $(|m\rangle, |n\rangle)$ effectively contributing, so that one needs to sum over all of them.

The first-order contribution depends on the model we consider. We give a full account of the PRBM case. We only give the main results for the RS model, since it essentially follows the same lines and was already partially discussed in [46].

### 5.2 PRBM model

#### 5.2.1 Order 1

For the PRBM model, the operator $\hat{M}$ of interest is the tight-binding Hamiltonian $\hat{H}$ defined in Sec. 2.1. The $2 \times 2$ submatrices of $H_{nm}$ contributing to first order Eq. (68) can be parametrized as

$$\begin{pmatrix} H_{mm} & H_{mn} \\ H_{mn}^* & H_{nn} \end{pmatrix} = \begin{pmatrix} \varepsilon + \Delta & re^{i\xi} \\ re^{-i\xi} & \varepsilon - \Delta \end{pmatrix}. \tag{69}$$

The average in Eq. (68) now runs over disorder realisations of parameters $\varepsilon$, $\Delta$, $r$ and $\xi$.

As explained in Sec. 2.1, entries $H_{mm}$ and $H_{nn}$ of the PRBM model are independent random real numbers with Gaussian distribution of variance 1. Off-diagonal entries $H_{mn}$ are complex random numbers, whose real and imaginary part are independent with Gaussian distribution of variance $\sigma_{nm}^2/2$, with $\sigma_{nm}$ given by (3). This means that $\varepsilon = \frac{1}{2}(H_{mm} + H_{nn})$ and $\Delta = \frac{1}{2}(H_{mm} - H_{nn})$ in (69) both have Gaussian distribution with variance $1/2$, while $\xi$ is uniformly distributed in $[0, 2\pi]$ and $r = \sqrt{|H_{nm}|^2} \in [0, \infty)$ is distributed with PDF $f_T(r)$ given by

$$f_T(r) = \frac{2r}{\sigma_{mn}^2} \exp\left(-\frac{r^2}{\sigma_{mn}^2}\right). \tag{70}$$

Eigenvectors $\left|\phi_{mn}^{\mu}\right\rangle$ with energy $E_{mn}^{\mu}$ of submatrices (69) can be expressed as

$$\left|\phi_{mn}^{+}\right\rangle = \cos\theta \, |m\rangle + e^{-i\xi} \sin\theta \, |n\rangle \,, \tag{71}$$

$$\left|\phi_{mn}^{-}\right\rangle = -e^{i\xi} \sin\theta \, |m\rangle + \cos\theta \, |n\rangle \,, \tag{72}$$

where angle $\theta$ is defined by

$$\tan\theta = -\frac{\Delta}{r} + \sqrt{1 + \frac{\Delta^2}{r^2}}. \tag{73}$$

The corresponding energy is

$$E_{mn}^{\mu} = \varepsilon + \mu\sqrt{r^2 + \Delta^2}. \tag{74}$$

The quantity of interest $\left|\left\langle k\middle|\phi_{mn}^{\mu}\right\rangle\right|^2$ then writes

$$\left|\left\langle k\middle|\phi_{mn}^{\mu}\right\rangle\right|^2 = 1 + \mu\cos\varphi_k \sin 2\theta \,, \tag{75}$$

with $\varphi_k = (m-n)k - \xi$. Performing the full calculation shows that the 1 in this expression is the 0th order contribution (this can be intuited by comparing this expression with the 0th order one). The order-1 contribution (68) then writes

$$\mathcal{I}^{(1)} = \left\langle \sum_{m<n} \frac{1}{N} \cos\varphi_k \cos\varphi_{k_0} \sin^2 2\theta \sum_{\mu=\pm 1} \delta(E - E_{mn}^{\mu}) \right\rangle. \tag{76}$$

Only $\varphi_k$ and $\varphi_{k_0}$ depend on $\xi$; averaging over it leads to

$$\mathcal{I}^{(1)}(E) = \sum_{m<n} \frac{1}{N} \cos([m-n][k-k_0]) \mathcal{A}_{mn}^{\mathrm{PRBM}}(E), \tag{77}$$

with

$$\mathcal{A}_{mn}^{\mathrm{PRBM}}(E) = \left\langle \frac{1}{2} \sin^2 2\theta \sum_{\mu=\pm 1} \delta(E - E_{mn}^{\mu}) \right\rangle. \tag{78}$$

The dependency of the above expression on $m$ and $n$ is via the parameter $r$, distributed according to Eq. (70). In particular, (78) only depends on the difference $|m-n|$. Moreover, in the periodic PRBM model we are considering, pair $(m, N-n)$ gives the same contribution as pair $(m,n)$ in Eq. (77) (the average (78) is taken over the same random realizations of parameters $r$, $\varepsilon$ and $\Delta$ for both pairs). As a consequence, the contrast up to order 1 writes

$$\Lambda_N(k, t \to \infty; E) = \sum_{n=1}^{N/2} \frac{\mathcal{A}_{n_0, n_0+n}^{\mathrm{PRBM}}(E)}{\rho(E)} \cos(n[k-k_0]). \tag{79}$$

We now find an explicit expression for $\mathcal{A}_{nm}^{\mathrm{PRBM}}(E)$. To do so, we use the fact that $\sin^2 2\theta = r^2/(r^2 + \Delta^2)$ and perform the remaining averages over $\varepsilon$, $\Delta$ and $r$ in Eq. (78). It gives

$$\mathcal{A}_{nm}^{\mathrm{PRBM}}(E) = \int_{-\infty}^{\infty} \frac{\mathrm{d}\Delta}{\sqrt{\pi}} e^{-\Delta^2} \int_{-\infty}^{\infty} \frac{\mathrm{d}\varepsilon}{\sqrt{\pi}} e^{-\varepsilon^2} \tag{80}$$

$$\times \int_0^{\infty} \mathrm{d}r \, \frac{2r}{\sigma_{mn}^2} e^{-\frac{r^2}{\sigma_{mn}^2}} \frac{r^2}{2(r^2 + \Delta^2)} \sum_{\mu=\pm} \delta(E - \varepsilon - \mu\sqrt{r^2 + \Delta^2}). \tag{81}$$

For $E = 0$, the integral (81) can be calculated explicitly, and for $b \to 0$ (where $\sigma_{mn}^2 \approx \frac{b(\pi/N)}{\sin \pi |n-m|/N} \ll 1$ for $m \neq n$) it gives at lowest order

$$\frac{\mathcal{A}_{mn}^{\mathrm{PRBM}}(E=0)}{\rho(E=0)} = \frac{\pi}{\sqrt{2}} \sigma_{mn} + \dots, \tag{82}$$

(we used the fact that $\rho(E)$ is given by Eq. (5) for $b \ll 1$). Finally, we find

$$\Lambda_N(k, t \to \infty, E=0) = \frac{b\pi}{\sqrt{2}} \sum_{n=1}^{N/2} \frac{(\pi/N) \cos(n[k-k_0])}{\sin(\pi n/N)}. \tag{83}$$

This result is checked in Fig. 5 (top) against numerics; the agreement is remarkable.

### 5.2.2 Asymptotic behavior of the peak height

At $k = k_0$, the contrast behaves following Eq. (56). In the regime of small parameter $b$, an expansion of the multifractal dimension $D_2$ was obtained in [50], using the same perturbative approach as above. At first order it reads $D_2 = b\pi/\sqrt{2}$. From Eq. (56) we get for $b \ll 1$

$$\Lambda_N(k_0, t \to \infty, E=0) \approx 1 - N^{-b\pi/\sqrt{2}} \sim \frac{b\pi}{\sqrt{2}} \ln N. \tag{84}$$

This expression coincides with the leading term of Eq. (83). Indeed, in the sum

$$\sum_{n=1}^{N/2} \frac{\pi}{N \sin(\pi n/N)} = \frac{\pi}{N} \sum_{n=1}^{N/2} \left( \frac{1}{\sin(\pi n/N)} - \frac{1}{\pi n/N} \right) + \sum_{n=1}^{N/2} \frac{1}{n}, \tag{85}$$

the first term is a Riemann sum that converges to the finite value $\ln(4/\pi)$, while the second term behaves asymptotically as $\sim \ln N$. Thus Eq. (83) at $k = k_0$ entails the asymptotic behavior Eq. (84) with the correct prefactor. This provides a check of Eq. (56) in the perturbation regime.

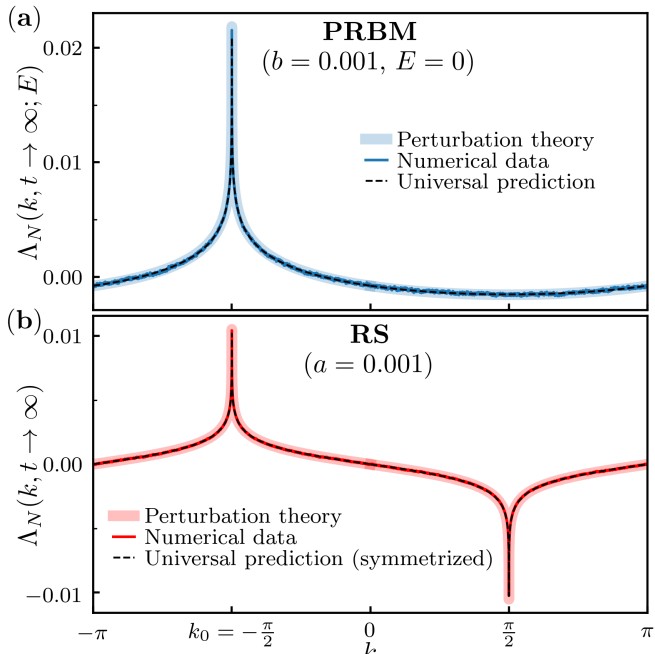

Figure 5: CFS contrast in the long-time limit and strong multifractal regime. (a) PRBM model for $b = 0.001$ ($N = 16384$, $n_d = 1125$ disorder realizations). (b) RS model for $a = 0.001$ ($N = 16384$, $n_d = 900$ disorder realizations). In both plots, thick solid lines are results from perturbation theory Eqs. (83) and (97), thin solid lines are numerical data (see Appendix B for details). For PRBM model, dashed black line is the universal prediction Eq. (54) (for $|k - k_0| \gg 1$) with $D_2 = 0$ and height adjusted to best fit the numerical data. For RS model, dashed black line is the symmetrized universal prediction Eq. (101) (see text), numerical data are averaged over $E$.

### 5.2.3 Expansion of the two-point correlator in direct space

The comparison of Eq. (79) with the universal analytical expression Eq. (52) suggests that $\mathcal{A}_{nm}^{\mathrm{PRBM}}(E)$ is equal up to order 1 to the two-point correlation function in direct space, that is,

$$\mathcal{B}_{nm}(E) = \Big\langle \sum_\alpha |\phi_\alpha(n)|^2 |\phi_\alpha(m)|^2 \delta(E - E_\alpha) \Big\rangle . \tag{86}$$

This can be shown directly as follows. As previously, we expand $\mathcal{B}_{nm}(E)$ as

$$\mathcal{B}_{nm}(E) = \mathcal{B}_{nm}^{(0)}(E) + \mathcal{B}_{nm}^{(1)}(E) + \dots . \tag{87}$$

Expression (86) at order 0 gives

$$\mathcal{B}_{nm}^{(0)}(E) = \Big\langle \sum_l |\langle n|l\rangle|^2 |\langle m|l\rangle|^2 \delta(E - E_l) \Big\rangle , \tag{88}$$

which vanishes for $n \neq m$. At order 1, using eigenstates (71)–(72) we find

$$
\begin{aligned}
\mathcal{B}_{nm}^{(1)}(E) &= \Big\langle \sum_{l<p} 2\sin^2\theta \cos^2\theta \, \delta_{nl}\delta_{mp} \sum_{\mu=\pm} \delta(E - E_{lp}^\mu) \Big\rangle \\
&= \Big\langle \frac{1}{2}\sin^2 2\theta \sum_{\mu=\pm} \delta(E - E_{mn}^\mu) \Big\rangle .
\end{aligned} \tag{89}
$$

This proves that $\mathcal{A}_{nm}^{\mathrm{PRBM}}(E) = \mathcal{B}_{nm}(E)$ up to order 1. In particular Eq. (79) becomes

$$\Lambda_N(k, t \to \infty; E) = \sum_{n=1}^{N/2} \langle |\phi_\alpha(n_0)|^2 |\phi_\alpha(n_0 + n)|^2 \rangle_E \cos(n[k - k_0]), \tag{90}$$

which is exactly the universal analytical expression Eq. (52).

## 5.3 RS model

We now apply the same method to determine the first order contribution $\mathcal{I}^{(1)}(E)$ for the RS model, which is unitary. We give the key points and main results. The interested reader should refer to the supplementary material of [46], in which more details are given.

The operator $\hat{M}$ of interest for the RS model is defined as $M_{nm} = U_{nm}e^{-i\pi a(1-1/N)}$, where $\hat{U}$ is the Floquet operator (9). This transformation only shifts the eigenvalues of $\hat{U}$ and has no physical consequences (in particular the multifractal dimensions remain unchanged). In the strong multifractal regime $a \ll 1$, the operator $\hat{M}$ in direct space writes

$$M_{nm} \simeq e^{i\varphi_n}\delta_{nm} - \frac{2i\pi a}{N}e^{i\varphi_n}\frac{1 - \delta_{nm}}{1 - e^{2\pi i(n-m)/N}}. \tag{91}$$

The term of order 0 is diagonal. At order 1, the $2 \times 2$ submatrices contributing to Eq. (68) read

$$\begin{pmatrix} M_{mm} & M_{mn} \\ M_{nm} & M_{nn} \end{pmatrix} = \begin{pmatrix} e^{i\varphi_m} & he^{i(\varphi_m + \xi)} \\ he^{i(\varphi_n - \xi)} & e^{i\varphi_n} \end{pmatrix}, \tag{92}$$

with

$$h = \frac{a\pi/N}{\sin\frac{(m-n)\pi}{N}} \quad \text{and} \quad \xi = \frac{\pi(m-n)}{N}. \tag{93}$$

These submatrices only depend on two independent random parameters $\varphi_m$ and $\varphi_n$, while the off-diagonal amplitudes $h$ are deterministic, unlike PRBM.

As previously, it is more convenient to introduce the random variables $\Delta = \frac{1}{2}(\varphi_m - \varphi_n)$ and $\varepsilon = \frac{1}{2}(\varphi_m + \varphi_n)$. Following the same lines, we find that the first order contribution can be written as

$$\mathcal{I}^{(1)}(E) = \sum_{n=1}^{N/2} \mathcal{A}_{n_0, n_0+n}^{\mathrm{RS}}(E) \cos(n[k - k_0]) - \sum_{n=1}^{N/2} \mathcal{A}_{n_0, n_0+n}^{\mathrm{RS}}(E) \cos\left(n[k + k_0 + \frac{2\pi}{N}]\right), \tag{94}$$

where

$$\frac{\mathcal{A}_{m,n}^{\mathrm{RS}}(E)}{\rho(E)} = \int_{-\pi/2}^{\pi/2} \frac{d\Delta}{\pi} \frac{h^2}{h^2 + \sin^2\Delta}, \tag{95}$$

does not depend on $E$. In the limit $a \to 0$ it gives

$$\frac{\mathcal{A}_{m,n}^{\mathrm{RS}}(E)}{\rho(E)} \approx h - \frac{h^3}{2} + \dots, \tag{96}$$

so that finally

$$\Lambda(k, t \to \infty, E) = a \sum_{n=1}^{N/2} \left[ \frac{(\pi/N)\cos(n[k - k_0])}{\sin(\pi n/N)} - \frac{(\pi/N)\cos\left(n[k + k_0 + \frac{2\pi}{N}]\right)}{\sin(\pi n/N)} \right], \tag{97}$$

which is independent of $E$. The first term describes the CFS peak, and is similar to the PRBM result (83). The second term describes an anti-CBS peak, that we will discuss in Section 5.4.3 below.

As before, we can show that $\mathcal{A}_{m,n}^{\text{RS}}(E)$ is nothing but the two-point correlation function in direct space (for $n \neq m$) up to order 1 of perturbation theory, that is,

$$\mathcal{A}_{m,n}^{\text{RS}}(E) \approx \langle \sum_\alpha |\phi_\alpha(n)|^2 |\phi_\alpha(m)|^2 \delta(E - E_\alpha) \rangle . \tag{98}$$

## 5.4 Comparison with numerics and universal predictions

### 5.4.1 Comparison with numerics

In Fig. 5 we display the results of our perturbation theory calculations for PRBM at $E = 0$ and for RS. Both reproduce very accurately the numerics in the strong multifractality limit.

### 5.4.2 Comparison with universal predictions

Leaving out the anti-CBS peak contribution in RS model for now, we see from Fig. 5 that both in PRBM and RS the CFS contrast in the long-time limit fully corroborates the universal analytical expression Eq. (52) (after pairing contributions $n$ and $-n$), that is

$$\Lambda(k, t \to \infty, E) = \sum_{n=1}^{N/2} \langle |\phi_\alpha(n_0)|^2 |\phi_\alpha(n_0 + n)|^2 \rangle_{E,n_0} \cos(n[k - k_0]). \tag{99}$$

Actually, at first order of perturbation theory these two models even have the same expression around the CFS peak

$$\Lambda(k, t \to \infty, E) \sim \sum_{n=1}^{N/2} \frac{(\pi/N) \cos(n[k - k_0])}{\sin(\pi n/N)} , \tag{100}$$

and only the prefactor differs. This is to be expected, since off-diagonal terms of PRBM ($r$ in (69)) and RS ($h$ in (92)) behave in the same way, namely $\sim \pi/N / \sin(\pi |n - m|/N)$.

### 5.4.3 Anti CBS-peak in RS model at small $a$

Let us now get back to the anti-CBS peak in the RS model. We see in Fig. 5 that this anti-peak is well captured by the perturbative expansion (97) while it is not present in the universal analytical prediction Eq. (54). However, we can adopt a phenomenological point of view and adapt the universal prediction : in order to take into account the anti-CBS peak, we propose that

$$\Lambda(k, t \to \infty, E) = A \sum_{n=1}^{N/2} \frac{\cos(n[k - k_0])}{n} + B \sum_{n=1}^{N/2} \frac{\cos\left(n[k + k_0 + \frac{2\pi}{N}]\right)}{n} , \tag{101}$$

where $A$ and $B$ are two fitting parameters. We then recover a very good agreement with numerical data (see Fig. 5b). This suggests that our approach missed some non vanishing contributions, probably due to a hidden symmetry inducing phase correlation of the eigenstates in direct space. This idea is corroborated by the observation (both from numerical data - not shown - and from perturbation theory) of an asymptotic symmetry verified by every single eigenstate in the perturbative regime, $|\phi_\alpha(k)|^2 + |\phi_\alpha(-k)|^2 \approx 2$. We will not dwell further on this peculiarity in the present work.

# 6   Summary and conclusion

We have studied CFS in critical disordered systems with multifractal eigenstates. We demonstrated that there exist two distinct dynamical regimes:

(i) When $t \ll \tau_H$, the CFS arises from the nonergodicity of the eigenstates. This regime corresponds to infinite system size and is relevant for most experimental situations. We recovered and demonstrated the numerical conjecture of [44] in the same limit: the CFS peak height asymptotically goes to $\chi = 1 - \frac{D_1}{d}$. We discovered that the CFS peak height actually reaches $\chi$ with a temporal power law related to the multifractal dimension $D_2$ (see Eq. (58)), and we gave a full description of the shape of the CFS peak: it gets smaller and smaller and the tail of the distribution decays with a power-law related to $D_2$ (see Eq. (63)).

(ii) When $t \gg \tau_H$, the CFS is caused by the system boundaries. The height of CFS peak goes to 1 with a finite-size correction related to multifractal dimension $D_2$, and the CFS shape decays as $N^{-D_2}$ elsewhere, the shape of the distribution being given by a system-size independent function (see Eq. (54)).

All our universal analytical predictions are verified very accurately on three critical disordered systems (PRBM, RS, 3DKR) in both strong and weak multifractal regimes. Moreover, for PRBM and RS models in the strong multifractality regime, we find that our universal predictions in the regime (ii) are exact at first order of perturbation theory.

These results, in particular (i), should be in reach of experiments, such as [38]. This opens the way to the first direct observation of a dynamical manifestation of multifractality in a critical disordered system.

## Acknowledgments.

OG wishes to thank MajuLab and CQT for their kind hospitality. This study has been supported through the EUR grant NanoX n° ANR-17-EURE-0009 in the framework of the "Programme des Investissements d'Avenir", and research funding Grants No. ANR-17-CE30-0024, ANR-18-CE30-0017 and ANR-19-CE30-0013. We thank Calcul en Midi-Pyrénées (CALMIP) for computational resources and assistance.

## A   Determination of the critical parameter $K_c$ of the unitary 3DKR

To determine the critical parameter $K_c$, at which the Anderson transition occurs in the 3DKR, we follow the lines of [66, 79], that we briefly recall here.

The one-parameter scaling theory predicts that at the Anderson transition diffusion is anomalous. Namely, starting from an initially fully localized wavefunction in direct space, i.e. $\langle \mathbf{p} | \psi(t = 0) \rangle = \delta(\mathbf{p})$, it predicts $\langle p^2 \rangle \propto t^{2/3}$.

From a numerical point of view, we simulate the dynamics of an initially localized wavepacket using the split-step scheme discussed in (B.16)-(B.17) below. We compute the standard deviation and plot $\langle p^2 \rangle \times t^{-2/3}$ as a function of time. At the critical point there should be no finite-size effect. The critical value of $K$ correspond to the flat curve in Fig. 6, yielding an estimate $K_c \approx 1.58$.

## B   Numerical methods

We give a detailed discussion of the different numerical procedures used in the article.

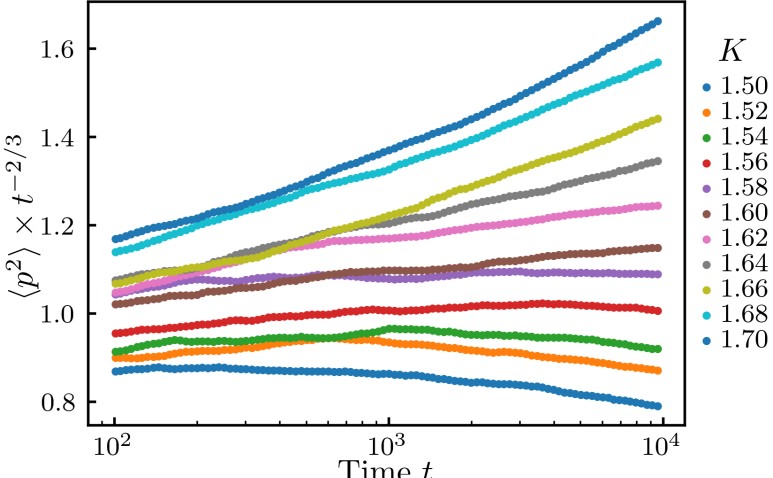

Figure 6: Determination of the critical kicking strength $K$ in the 3DKR model. System size is $N = 128$ and number of disorder realizations $n_d = 179$. $\langle p^2 \rangle$ is the momentum variance of an initially fully localized wavefunction. Symbols are numerical data. For the value $K = 1.58$, the curve $\langle p^2 \rangle \times t^{-2/3}$ is flat, indicating the critical point of the Anderson transition (see text).

## B.1  PRBM

### B.1.1  Energy filtering procedure

In order to evaluate $n(\mathbf{k}, t; E)$ defined in Eq. (34), we use a filtering technique introduced in [44]. Let $E_0$ be the targeted energy; the idea is to replace the initial state $|\psi_0\rangle = |k_0\rangle / \sqrt{N}$ by a Gaussian-filtered plane wave around $E_0$

$$|\psi_0\rangle = \frac{1}{(\sigma^2 \pi)^{1/4}} \exp\left(-\frac{(E_0 - \hat{H})^2}{2\sigma^2}\right) |k_0\rangle / \sqrt{N}, \tag{B.1}$$

where $\sigma$ is the width of the energy filter. The filtered scattering probability can be written as

$$n_{\text{fil}}(k, t; E_0) = \frac{1}{N} \Big\langle \sum_{\alpha, \beta} e^{-i\omega_{\alpha\beta}t} \frac{1}{\sigma \sqrt{\pi}} \exp\left(-\frac{(E_0 - \omega_\alpha)^2}{2\sigma^2}\right) \tag{B.2}$$

$$\times \exp\left(-\frac{(E_0 - \omega_\beta)^2}{2\sigma^2}\right) \phi_\alpha(k)\phi_\alpha^\star(k_0)\phi_\beta(k_0)\phi_\beta^\star(k) \Big\rangle, \tag{B.3}$$

or equivalently

$$n_{\text{fil}}(k, t; E_0) = \frac{1}{N} \int dE \int d\omega\, e^{-i\omega t} \Big\langle \frac{1}{\sigma \sqrt{\pi}} \exp\left(-\frac{(E_0 - E)^2}{\sigma^2}\right) \sum_{\alpha, \beta} \delta(\omega - \omega_{\alpha\beta}) \tag{B.4}$$

$$\times \exp\left(-\frac{\omega^2}{4\sigma^2}\right) \phi_\alpha(k)\phi_\alpha^\star(k_0)\phi_\beta(k_0)\phi_\beta^\star(k) \Big\rangle. \tag{B.5}$$

We see that $n_{\text{fil}}(k, t; E_0)/\rho(E_0)$ is not much different from $n(k, t; E_0)$ in Eq. (34), provided $\sigma$ is sufficiently small (compared with the DOS variation), because

$$\lim_{\sigma \to 0} \frac{1}{\sigma \sqrt{\pi}} \exp\left(-\frac{(E_0 - E)^2}{\sigma^2}\right) = \delta(E - E_0). \tag{B.6}$$

One noticeable difference however is the term $\exp\left(-\omega^2/4\sigma^2\right)$, that acts as a high energy cut-off in the filtered dynamics. Consequently, $n_{\text{fil}}(k,t;E_0)$ is coarse-grained over a time scale $\sim 1/\sigma$. In particular, simulating times shorter than $1/\sigma$ is not relevant.

In practice, eigenstate properties can be considered roughly constant in an energy window where the DOS (5) does not vary much. We choose

$$\sigma = \frac{1}{8}\max(\sqrt{\pi b}, 1). \tag{B.7}$$

For the values presented in the article ($b = 0.05, 0.1, 0.3$) the corresponding time scale $1/\sigma$ is of the order of 10. Note that data presented in Fig. 4 are additionally averaged on a timescale $\Delta t \geq 1/b\sigma$ for the sake of clarity.

The classical contribution, with filtered initial state, should write

$$n_{\text{class,fil}}(k;E_0) = \int dE\, \rho(E) \frac{1}{\sigma\sqrt{\pi}} \exp\left(-\frac{(E_0 - E)^2}{\sigma^2}\right) \frac{A(k,E)}{\rho(E)} \frac{A(k_0,E)}{\rho(E)}. \tag{B.8}$$

Again, we see that $n_{\text{class,fil}}(k,t;E_0)/\rho(E_0)$ is not much different from $n(k,t;E_0)$ in Eq. (36), provided that $\sigma$ is sufficiently small (compared with the DOS variation). Under the diagonal approximation ($A(k,E) = \rho(E)$), it becomes

$$n_{\text{class,fil}}(k;E_0) = \int dE\, \rho(E) \frac{1}{\sigma\sqrt{\pi}} \exp\left(-\frac{(E_0 - E)^2}{\sigma^2}\right) = \left\langle \frac{1}{N}\sum_\alpha \frac{1}{\sigma\sqrt{\pi}} \exp\left(-\frac{(E_0 - \omega_\alpha)^2}{\sigma^2}\right) \right\rangle, \tag{B.9}$$

where we used the definition (4) of $\rho(E)$.

The numerical contrast is thus finally defined as

$$\Lambda(k,t;E_0) = \frac{n_{\text{fil}}(k,t;E_0)}{\left\langle \frac{1}{N}\sum_\alpha \frac{1}{\sigma\sqrt{\pi}} \exp\left(-\frac{(E_0 - \omega_\alpha)^2}{\sigma^2}\right) \right\rangle} - 1, \tag{B.10}$$

and is actually independent of the choice of normalization for the energy filter because both $n_{\text{fil}}(k,t;E_0)$ and $n_{\text{class,fil}}(k;E_0)$ are proportional to $\frac{1}{\sigma\sqrt{\pi}}$.

### B.1.2 Infinite system size limit ($t \ll \tau_H$)

To evaluate the filtered contrast Eq. (B.10) in the regime $t \ll \tau_H$, we diagonalize PRBM matrices of size $N$ in an energy window $[-\sigma, \sigma]$ (this roughly corresponds to 1/4 of the eigenstates of the system) and expand the filtered time propagator over the eigenstates in the reciprocal space.

Combining conditions to reach the regime $t \ll \tau_H$, and the one coming from the filter (see below (B.6)), we get that the relevant time must verify (for small $b$)

$$\frac{1}{\sigma} \leq t \ll N. \tag{B.11}$$

We checked that the upper bound of this inequality was met by verifying that the CFS contrast was independent of the system size $N$, and that the filtered form factor Eq. (28), applying the same substitution as for the filtered contrast, directly computed from the knowledge of eigenvalues, for different times, was stationary. Note that this condition is a bit stronger than for the RS model, because here we use exact diagonalization (of a non-sparse matrix) to compute the dynamics, which limits us to system sizes about 10 times smaller than the ones simulated with the RS model using the split-step scheme.

The numbers of disorder realizations are given in Table 2.

Table 2: Number of numerical disorder realizations $n_d$ used to average statistical properties of the PRBM model, for different system sizes $N$.

| $N$ | 512 | 1024 | 2048 | 4096 | 8192 | 16384 |
|---|---|---|---|---|---|---|
| $n_d$ | 36000 | 18000 | 9000 | 4500 | 2160 | 1125 |

### B.1.3 Long-time limit ($t \gg \tau_H$)

To compute long-time dynamics, we use the identity (51). We express eigenstates in reciprocal space. We use the same number of disorder realizations as in Table 2.

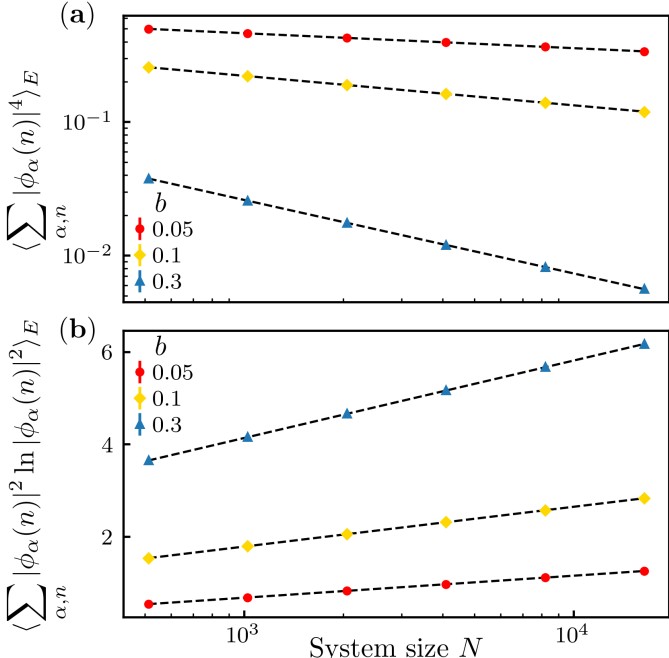

Figure 7: Determination of $D_1$ and $D_2$ in the PRBM model ($E = 0$) by finite-size scaling of moments $I_q(E)$ Eq. (1). (a) Determination of $D_2$. Symbols are numerical data, with error bars smaller than symbol size. Dashed lines are two-parameter fits $y = AN^{-D_2}$ (see Eq. (1)). (b) Determination of $D_1$. Symbols are numerical data, with error bar smaller than symbols. Dashed lines are two-parameter fits $y = B + N \ln D_1$ (see Eq. (31)). Numbers of disorder realizations are given in Table 2. Corresponding values of $D_1$ and $D_2$ are given in Table 3.

### B.1.4 Filtered multifractal properties

Multifractal dimensions are determined by filtering the finite-size scaling laws (1) and (31) of the moments $I_q(E)$. We express the eigenstates in the direct basis, then compute Eqs. (1) and (31) for different system sizes $N$ and average the results over $n_d$ different disorder realizations (see Table 2). Finally, we fit the averaged moments *vs* system size $N$ to obtain $D_1$ and $D_2$ (see Fig. 7). The results are given in Table 3.

Table 3: Numerically determined multifractal dimensions for the PRBM model ($E = 0$). Errors are always smaller than $10^{-6}$. $1 - \chi_{\text{num}}$ is given to test the validity of Eq. (30), with $\chi_{\text{num}}$ numerically determined by computing the form factor from eigenvalues, see Eqs. (28) and (29), in the same temporal interval than the CFS contrast, where it is constant and equal to the compressibility.

| $b$ | 0.05 | 0.1 | 0.3 |
|---|---|---|---|
| $D_1$ | 0.207 | 0.375 | 0.729 |
| $1 - \chi_{\text{num}}$ | 0.201 | 0.372 | 0.727 |
| $D_2$ | 0.112 | 0.221 | 0.551 |

### B.1.5    Spectral function

In the main text, we show that under the diagonal approximation the spectral function $A(k, E)$ does not depend on $k$ and is equal to $\rho(E)$, see Eq. (39). Here we verify explicitly numerically the validity of this approximation in PRBM. The numerical spectral function is defined via the above filtering technique, as

$$A(k, E) = \left\langle \frac{1}{N} \sum_\alpha \frac{1}{\sigma \sqrt{\pi}} \exp\left(-\frac{(E - \omega_\alpha)^2}{\sigma^2}\right) |\phi_\alpha(k)|^2 \right\rangle. \tag{B.12}$$

The density of states at energy $E$ is directly computed by counting the number of states in an interval of width $2\sigma$ around $E$. As shown in Fig. 8, we find a very good agreement of Eq. (39) with numerics, for different values of $E$ and different parameters $b$. This supports the validity of the diagonal approximation for the calculation of the classical background for the CFS peak.

### B.2    RS model

As discussed in the main text, in the RS model, the CFS contrast is independent of the mean energy $E$. In practice, we therefore compute the integrated probability $n(\mathbf{k}, t)$ defined in Eq. (34). The corresponding contrast is given by

$$\Lambda_N(\mathbf{k}, t) = n(\mathbf{k}, t) - 1. \tag{B.13}$$

It can be seen as the average of the energy-dependent contrast $\Lambda_N(\mathbf{k}, t; E)$ over all (equally contributing) energies, since

$$\begin{aligned}
\Lambda_N(\mathbf{k}, t) &= \int_0^{2\pi} \rho(E) n(\mathbf{k}, t; E) \, dE - 1 \\
&= \frac{1}{2\pi} \int_0^{2\pi} \Lambda_N(\mathbf{k}, t; E) \, dE.
\end{aligned} \tag{B.14}$$

### B.2.1    Infinite system size limit ($t \ll \tau_H$)

We recall that the Floquet operator of the RS model is the product of two operators,

$$\hat{U} = e^{-i\phi_{\hat{p}}} e^{-ia\hat{x}}, \tag{B.15}$$

where phases $\phi_{\hat{p}}$ are randomly generated in the interval $[0, 2\pi[$. The first operator represents kinetic energy during the free propagation and is diagonal in $p$ space. The second one represents the kick and is diagonal in $x$ space.

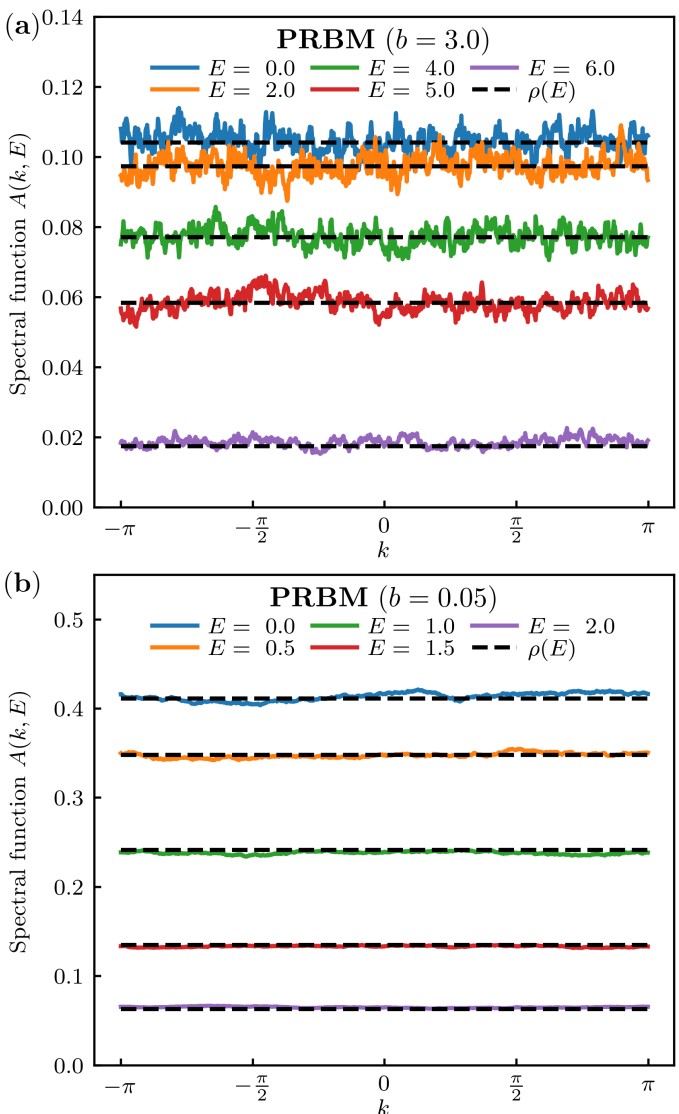

Figure 8: Spectral function $A(k, E)$ in PRBM for various values of $E$. Parameters are $N = 1024$, $n_d = 10$ disorder realizations. Solid lines are numerical data for the spectral function, dashed lines are numerical data for the density of state.

We use a grid of size $N$ (even) with positions evenly spaced in the interval $[0, 2\pi[$, $x_k = 2\pi k/N$, with $k$ integer. The corresponding grid in momentum space is $p = -N/2 + 1, \ldots, N/2$.

A wavefunction $\psi$ is initially prepared in a single position state around $x_0 = \pi/2$. The propagation scheme over one period is then achieved by applying twice a Fast Fourier Transform (FFT) algorithm, in the spirit of the split-step method

$$\psi(p, t = 0^+) = \mathrm{FFT}[e^{-iax_n}\psi(x_n, t = 0)], \tag{B.16}$$

$$\psi(x_n, t = 1) = \mathrm{FFT}^{-1}[e^{-i\phi_p}\psi(p, t = 0^+)]. \tag{B.17}$$

This method is particularly efficient and makes it possible to simulate very large system sizes, up to $N = 131072$, as in Fig. 4. To ensure that the condition $t \ll \tau_H = N$ is met, we checked that the CFS contrast is size-independent.

### B.2.2  Long-time limit

To compute long-time dynamics, we use the identity (51). We compute and diagonalize the Floquet operator and express the eigenstates in the reciprocal basis (here the $x$ basis). The number of disorder realizations for each system size is given in Table 4.

Table 4: Number of numerical disorder realizations $n_d$ used to average statistical properties of the RS model, for different system sizes $N$.

| $N$ | 512 | 1024 | 2048 | 4096 | 8192 | 16384 |
|---|---|---|---|---|---|---|
| $n_d$ | 28800 | 14400 | 7200 | 3600 | 1800 | 900 |

Diagonalizing the matrices is more computationally demanding than naive time propagation at long time $t \gg \tau_H$ (which scales as $\sim N$ for each time step). However, results are more reliable because of the oscillatory nature of the large-time behavior in the RS model. Indeed, the form factor of the RS model is given by [59]

$$K(t) = \frac{(1-a)^2(\kappa t)^2}{a^2(1-\cos\kappa t)^2 + (a\sin\kappa t + (1-a)\kappa t)^2}\,, \tag{B.18}$$

with $\kappa = 2\pi a/N$, and has the following asymptotic expansion

$$K(t) \underset{t \gg N/a}{\approx} 1 - \frac{2a\sin(\kappa t)}{(1-a)\kappa t}\,. \tag{B.19}$$

Because of Eq. (49), this slow algebraic and oscillatory convergence to its limiting value also manifests itself in the CFS contrast, which significantly complicates the numerical determination of the asymptotic contrast.

### B.2.3  Multifractal dimensions

Multifractal dimensions are determined using finite-size scaling laws (1) and (31) of the moments $I_q(E)$. However, as $I_q(E)$ (and $D_q$) do not depend on $E$ for RS, we compute averaged moments $\overline{I_q}$ over all quasi-energies $E$

$$\overline{I_q} = \frac{1}{2\pi}\int_0^{2\pi} dE\, I_q(E) = \langle \frac{1}{N}\sum_{\alpha,n}|\phi_\alpha(n)|^{2q}\rangle\,. \tag{B.20}$$

We compute and diagonalize the Floquet operator and express the eigenstates in the direct basis (momentum basis). Then we compute Eqs. (1) and (31) for different system sizes $N$ and average the results over $n_d$ different disorder realizations (see Table 4). Finally, we fit the averaged moments *vs* system size $N$ to obtain $D_1$ and $D_2$ (see Fig. 9). The results are given in Table 5.

### B.3  3DKR

Similarly to the RS model, the 3DKR is a Floquet system, whose eigenstate properties do not depend on quasienergy. We thus compute the integrated contrast (B.13).

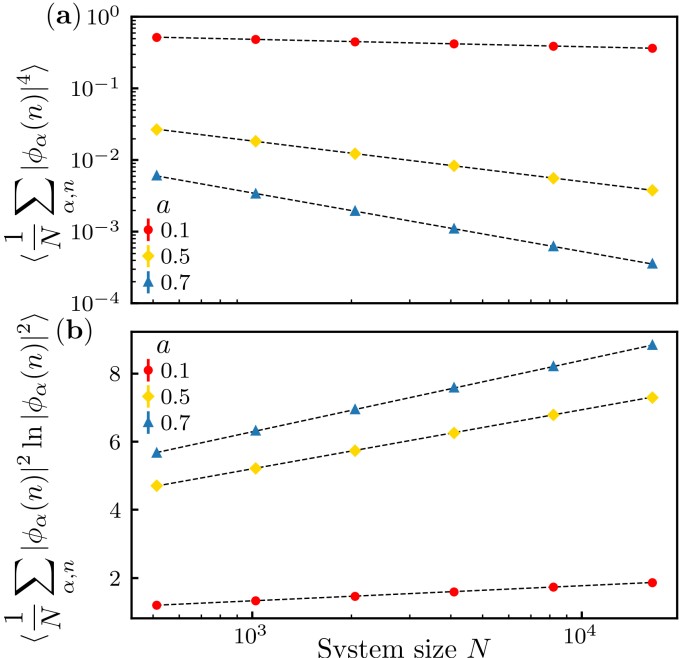

Figure 9: Determination of $D_1$ and $D_2$ in the RS model by finite-size scaling of moments $\overline{I_q}$ given by Eq. (B.20). (a) Determination of $D_2$. Symbols are numerical data, with error bars smaller than symbol sizes. Dashed lines are two-parameter fits $y = AN^{-D_2}$. (b) Determination of $D_1$. Symbols are numerical data, with error bars smaller than symbol sizes. Dashed lines are two-parameter fits $y = B + N \ln D_1$. Numbers of disorder realizations are given in Table 4. Corresponding values of $D_1$ and $D_2$ are given in Table 5.

### B.3.1 Infinite system size limit ($t \ll \tau_H$)

We use the exact same method as for the RS model, based on the propagation of wavefunctions with the split-step scheme Eqs. (B.16)-(B.17), except that we now use a 3d grid. To ensure that the condition $t \ll \tau_H = N^3$ is met, we checked that the CFS contrast is size-independent.

### B.3.2 long-time limit ($t \gg \tau_H$)

Unlike for RS model, to access the long-time dynamics we used temporal propagation of the wavefunction up to time $t \sim \tau_H$. We observed that beyond $t > 2.5N^3$, the contrast reaches a stationary value; we thus averaged the CFS contrast in the temporal window $2.5 < t/N^3 < 3$ for different system sizes.

Note that the computational time to reach this regime scales as $\sim N^3 \times N$ with system size $N$, which is why we limited ourselves to $N = 128$ ($N = 256$ would for instance require to reach $50 \times 10^6$ kicks with a system of $256^3$ points).

Table 5: Numerically determined multifractal dimensions for the RS model. Errors are negligible (always smaller than $10^{-6}$). $1 - \chi_{\text{th}}$ is given to test the validity of Eq. (30) (with $\chi_{\text{th}} = (1-a)^2$ (see [59] or Eq. (B.18) for $t \to 0$).

| $a$ | 0.1 | 0.5 | 0.7 |
|---|---|---|---|
| $D_1$ | 0.192 | 0.753 | 0.910 |
| $1 - \chi_{\text{th}}$ | 0.190 | 0.750 | 0.910 |
| $D_2$ | 0.103 | 0.566 | 0.820 |

# C  Fourier transform and normalization conventions

**Closure relations**

$$\mathbb{I} = \sum_{\mathbf{n}} |\mathbf{n}\rangle\langle\mathbf{n}| \,, \tag{C.1}$$

$$\mathbb{I} = \frac{1}{N^d} \sum_{\mathbf{k}} |\mathbf{k}\rangle\langle\mathbf{k}| \xrightarrow[N\to\infty]{} \int \frac{d\mathbf{k}}{(2\pi)^d} |\mathbf{k}\rangle\langle\mathbf{k}| \,. \tag{C.2}$$

**Orthonormalization**

$$\langle\mathbf{n}|\mathbf{n}'\rangle = \delta_{\mathbf{n}\mathbf{n}'} \,, \tag{C.3}$$

$$\langle\mathbf{k}|\mathbf{k}'\rangle = N^d \delta_{\mathbf{k}\mathbf{k}'} \xrightarrow[N\to\infty]{} (2\pi)^d \delta(\mathbf{k}-\mathbf{k}') \,. \tag{C.4}$$

**Fourier transform**

$$|\mathbf{k}\rangle = \sum_{\mathbf{n}} e^{-i\mathbf{k}\cdot\mathbf{n}} |\mathbf{n}\rangle \,, \tag{C.5}$$

$$|\mathbf{n}\rangle = \frac{1}{N^d} \sum_{\mathbf{k}} e^{i\mathbf{k}\cdot\mathbf{n}} |\mathbf{k}\rangle \xrightarrow[N\to\infty]{} \int \frac{d\mathbf{k}}{(2\pi)^d} e^{i\mathbf{k}\cdot\mathbf{n}} |\mathbf{k}\rangle \,, \tag{C.6}$$

$$\langle\mathbf{n}|\mathbf{k}\rangle = e^{-i\mathbf{k}\cdot\mathbf{n}} \,. \tag{C.7}$$

**Eigenfunctions**

$$\sum_{\alpha} |\phi_\alpha\rangle\langle\phi_\alpha| = \mathbb{I}, \tag{C.8}$$

$$\sum_{\mathbf{n}} |\phi_\alpha(\mathbf{n})|^2 = 1 \,, \tag{C.9}$$

$$\frac{1}{N^d} \sum_{\mathbf{k}} |\phi_\alpha(\mathbf{k})|^2 = 1 \,, \tag{C.10}$$

$$\frac{1}{N^d} \sum_{\alpha} |\phi_\alpha(\mathbf{k})|^2 = 1 \,. \tag{C.11}$$

**Spectral function**

$$A(\mathbf{k}, E) = \frac{1}{N^d} \Big\langle \sum_{\alpha} |\phi_\alpha(\mathbf{k})|^2 \delta(E - \omega_\alpha) \Big\rangle \,, \tag{C.12}$$

$$\frac{1}{N^d}\sum_{\mathbf{k}} A(\mathbf{k}, E) = \frac{1}{N^d}\left\langle \sum_{\alpha} \delta(E - \omega_\alpha)\right\rangle = \rho(E), \tag{C.13}$$

$$\int dE\, A(\mathbf{k}, E) = \frac{1}{N^d}\left\langle \sum_{\alpha} |\phi_\alpha(\mathbf{k})|^2 \right\rangle = 1. \tag{C.14}$$

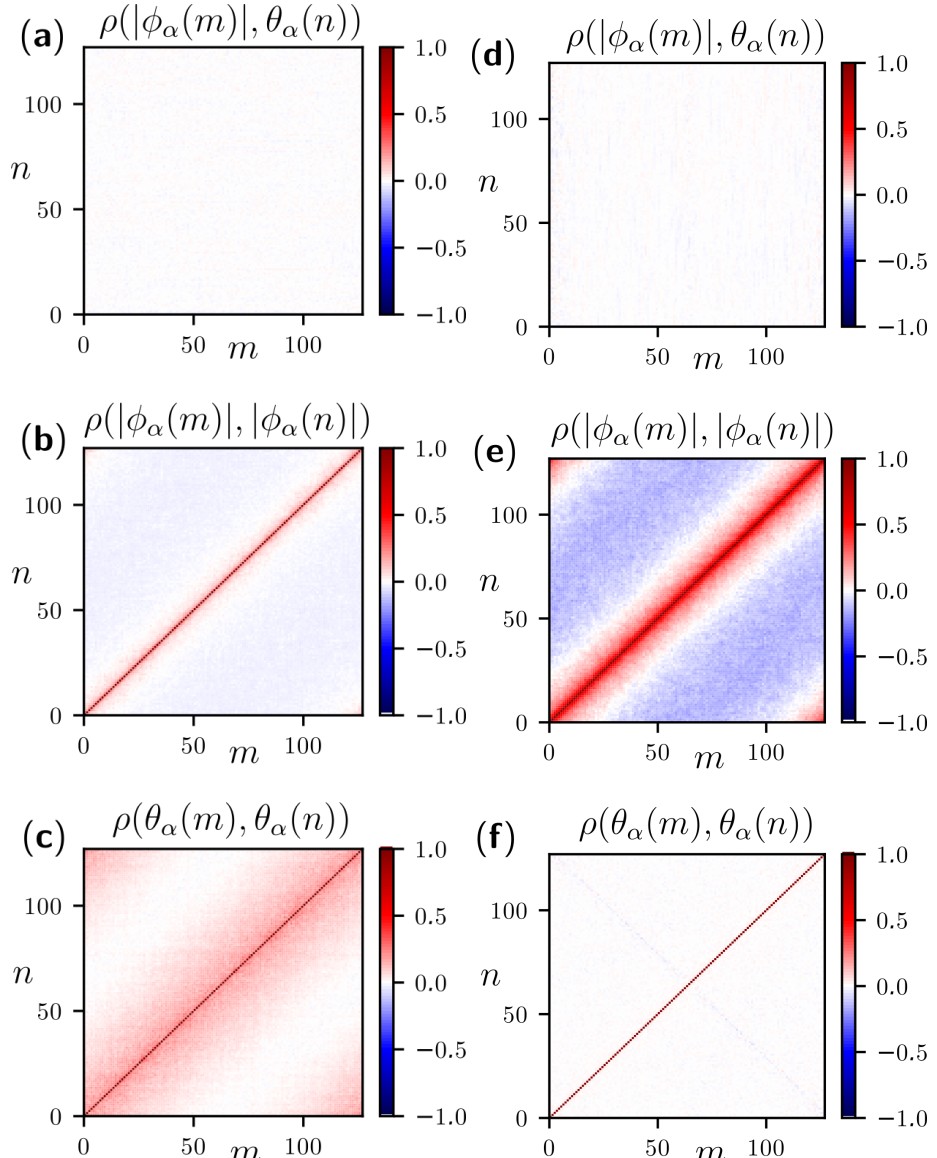

Figure 10: Correlations $\rho(X, Y) = (\langle XY\rangle - \langle X\rangle\langle Y\rangle)/(\sigma_X \sigma_Y)$ between norm $|\phi_\alpha(m)|$ and phase $\theta_\alpha(n)$ of a same eigenvector $\phi_\alpha = |\phi_\alpha|\exp(i\theta_\alpha)$ of the RS model, evaluated at different momenta $(m, n)$: (a,d) norm-phase correlation, (b,e) norm-norm correlation, (c,f) phase-phase correlation. Panels (a-c) correspond to the strong multifractal regime $a = 0.1$, panels (d-f) to the weak multifractal regime $a = 0.9$ (d-f). Matrix size is $N = 128$ and average is taken over disorder (100 realizations) and eigenvectors.

# D Incoherent background for PRBM

We have

$$A(k;E) = \frac{1}{N} \left\langle \sum_\alpha |\phi_\alpha(k)|^2 \delta(E - \omega_\alpha) \right\rangle \tag{D.1}$$

$$= \frac{1}{N} \left\langle \langle k| \sum_\alpha |\phi_\alpha\rangle\langle\phi_\alpha| \, \delta(E - \omega_\alpha) |k\rangle \right\rangle. \tag{D.2}$$

Using our temporal Fourier transform convention (18), this gives

$$A(k;t) = \frac{1}{2\pi N} \left\langle \langle k| \sum_\alpha |\phi_\alpha\rangle\langle\phi_\alpha| e^{-i\omega_\alpha t} |k\rangle \right\rangle \tag{D.3}$$

$$= \frac{1}{N} \left\langle \langle k| \hat{U}^t |k\rangle \right\rangle, \tag{D.4}$$

where we have used the eigenvalue-eigenvector decomposition

$$\hat{U} = \exp(-i\hat{H}) = \sum_\alpha |\phi_\alpha\rangle\langle\phi_\alpha| e^{-i\omega_\alpha}. \tag{D.5}$$

Expanding the exponential $\exp(-i\hat{H}t)$ into a series, Eq. (D.4) becomes

$$A(k;t) = \frac{1}{N} \sum_{n=0}^\infty \frac{(-it)^n}{n!} \left\langle \langle k| \hat{H}^n |k\rangle \right\rangle. \tag{D.6}$$

Changing to the direct basis, one has, using the closure relation (C.1),

$$\langle k| \hat{H}^n |k\rangle = \sum_{i,j} \langle k|i\rangle \langle i| \hat{H}^n |j\rangle \langle j|k\rangle. \tag{D.7}$$

The $N \times N$ Hamiltonian matrix in direct space has independent (up to Hermiticity) Gaussian entries $H_{ij} = \langle i| \hat{H} |j\rangle$. Calculating (D.6) requires to determine the averages of quantities

$$\left\langle \langle i| \hat{H}^n |j\rangle \right\rangle = \sum_{i_1,\ldots,i_n} \left\langle H_{ii_1} H_{i_1 i_2} \ldots H_{i_{n-1} i_n} H_{i_n j} \right\rangle. \tag{D.8}$$

The vector $(H_{11}, \text{Re}(H_{12}), \text{Im}(H_{12}), \ldots, H_{NN})$ is a multivariate centered Gaussian. Each moment in (D.8) can be calculated using Wick's theorem: moments of odd order vanish, and moments $\langle x_{a_1} \ldots x_{a_{2p}} \rangle$ are given by the sum over all possible pairings of the set $\{1, \ldots, 2p\}$. Because of independence of matrix elements, only entries $H_{ab}$ and $H_{ba}$ are non-independent; thus the only nonvanishing two-point correlators are either of the form $\langle H_{ab} H_{ba} \rangle$ or of the form $\langle H_{ab}^2 \rangle$ (possibly with $a = b$). That is, any given index in (D.8) must appear an even number of times. But all indices $i_1, i_2, \ldots, i_n$ do already appear in pairs. Therefore the two remaining indices $i$ and $j$ must be equal, otherwise at least one index would appear an odd number of times.

As a consequence, all terms with $i \neq j$ vanish in Eq. (D.8). Therefore, upon average, (D.7) yields for any fixed $k$

$$\left\langle \langle k| \hat{H}^n |k\rangle \right\rangle = \sum_i |\langle k|i\rangle|^2 \langle i| \hat{H}^n |i\rangle = \sum_i \langle i| \hat{H}^n |i\rangle, \tag{D.9}$$

using the normalization $|\langle k|i\rangle|^2 = 1$ (see Appendix C). Since each $\left\langle \langle k| \hat{H}^n |k\rangle \right\rangle$ is independent of $k$, so is $A(k;t)$ in Eq. (D.6). The identity $A(\mathbf{k};E) = \rho(E)$ then ensues from the normalization condition (C.13) of the spectral function.

# E Decorrelation between norms and phases

In the main text we perform our calculations under the approximation that norms and phases of random wavefunctions are uncorrelated, an assumption which is quite usual in random matrix theory. In order to assess this assumption, we illustrate it below in the case of the RS model and for different values of $D_q$. As shown in Fig. 10, norms and phases are indeed uncorrelated in the RS model.

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
