# Peer review of "Coherent forward scattering as a robust probe of multifractality in critical disordered media"

_SciPost Physics, doi:SciPost Phys. 14, 057 (2023)_

## Round 1 · Referee Report · Vladimir Kravtsov (Referee 1) · 2022-11-7

Strengths

1- Very well written paper 2-Detailed derivation of main results 3-Comparison of numerics on three different models, both Hamiltonian (PLBRM) and Floquet (kick-rotor models), confirms the main results

Weaknesses

1- significant overlap with the earlier short paper of the same authors Ref.[46]. For an expert the derivations of results in this paper is sufficient to understand their validity and significance. The present manuscript does not add much to understanding of the phenomenon.

Report

This is a very comprehensive and well written paper, the extension of a short paper of the same authors Ref.[46]. Besides a detailed derivation of all the main results (presented partially in Ref.[46]) this paper contains not only the dynamics of the peak in CFS but also a new results about the form of the wings of the peak and, especially, much more numerics on three different models that confirms the analytical results.

In view of a great current interest to non-ergodicity of extended states in quantum disordered systems with and without interaction, this study (as well as Ref.[46]) is a significant progress in the field with a potential to observation of the effects in real experiments.

I do not have doubts about validity of the results and their significance. The only reservation I have is that the most interesting and fundamental findings are already communicated by authors in Ref.[46].

On the other hand, many of the important papers on the subject of non-ergodicity of extended states in disordered quantum systems are published in SciPost Physics. So, this paper may be a good addition to already existent collection in SciPost Physics. I think it will have an impact just because it is easy to read, also for non-specialists.

In view of this I recommend to publish the paper in its present form.

---

## Round 1 · Referee Report · Anonymous (Referee 2) · 2022-11-20

Strengths

1-interesting details to previous publication
2-new simulation of system, with perspective for experimental realization
3-well written

Weaknesses

1-main results earlier published

Report

In their manuscript the authors extend previous studies of the coherent forward scattering peak in disordered single particle systems from the Anderson localized regime to the critical state at an Anderson localization transition. They identify two distinct dynamical regimes, applicable in the (non-commuting) limits of large systems sizes and long times, and give general arguments relating peak-height and -shape to multifractal properties of the critical system. In the infinite size system the forward peak saturates to the level compressibility (as previously conjectured in [44] and also demonstrated in [46]), while in the opposite limit (N finite, t \to \infty) the peak height is modified from random matrix prediction by a finite size correction that depends on the multifractal dimension D_2 of critical wave functions (as also previously reported in [46]). They verify predictions against three models for critical disordered systems, finding good agreement.

I think the manuscript reports on interesting work. It is well written, provides interesting details to their previous Letter, Ref. [46], and extends numerical studies in the latter to two additional models for the Anderson localization transition. Specifically, the detailed numerical analysis of the 3d kicked rotor seems interesting for possible future experiments. I, therefore, support publication of the manuscript in its present form. A minor remark is that I would appreciate if the authors could comment on the statistics of the forward peak? I.e. how do fluctuations of the peak height in the critical state compare to those in the Anderson localized regime?

---

## Editorial Decision

published